



# Diagnosis of future changes in hydrology for a Canadian Rocky Mountain headwater basin

Xing Fang and John W. Pomeroy

[1]Centre for Hydrology, University of Saskatchewan, Saskatoon, S7N 1K2, Canada

5   *Correspondence to*: Xing Fang (xing.fang@usask.ca)

**Abstract.** Climate change is anticipated to have impacts on the water resources of the Saskatchewan River, which originates in the Canadian Rocky Mountains.  To better understand the climate change impacts in the mountain headwaters of this basin, a physically based hydrological model was developed for this basin using the Cold Regions Hydrological Modelling platform (CRHM) for Marmot Creek Research Basin (~9.4 km$^2$), located in the Front Ranges of the Canadian Rocky Mountain.  Marmot 10   Creek is composed of ecozones ranging from montane forests to alpine tundra and alpine exposed rock and includes both large and small clearcuts.  The model included blowing and intercepted snow redistribution, sublimation, energy-balance snowmelt, slope and canopy effects on melt, Penman-Monteith evapotranspiration, infiltration to frozen and unfrozen soils, hillslope hydrology, streamflow routing and groundwater components and was parameterised without calibration from streamflow. Near-surface outputs from the 4-km Weather Research and Forecasting (WRF) model were bias corrected using the quantile 15   delta mapping method with respect to meteorological data from five stations located from montane forest to mountaintop during October 2005-September 2013.  The bias corrected WRF outputs during current period (CTRL, 2005-2013) and future period (PGW, 2091-2099) were used to drive model simulations to assess changes in Marmot Creek's hydrology.  Under a "business as usual" forcing scenario: representative concentration pathway 8.5 (RCP8.5) in PGW, the basin warms up by 4.7 °C and receives 16% more precipitation, which leads to a 40 mm decline in seasonal peak snowpack, 84 mm decrease in 20   snowmelt volume, 0.2 mm day$^{-1}$ slower melt rate, and 49 days shorter snowcover duration.  The alpine snow season will be shortened by almost one and half month, but at some lower elevations there are large decreases in peak snowpack (~45%) as well as a shorter snow season.  Losses of peak snowpack will be much greater in clearcuts than under forest canopies. In alpine and treeline ecozones blowing snow transport and sublimation will be suppressed by higher threshold wind speeds for transport, in forest canopies sublimation losses from intercepted snow will decrease due to faster unloading and drip, and for 25   all ecozones, evapotranspiration will increase due to longer snow-free seasons and more rainfall.  Runoff will begin earlier in all ecozones, but, as result of variability in surface and subsurface hydrology, forested and alpine ecozones generate larger runoff volumes, ranging from 12% to 25%, whereas the treeline ecozone has a small (2%) decrease in runoff volume due to decreased melt volumes from smaller snowdrifts.  The shift in timing in streamflow is notable, with 236% higher flows in spring months and 12% lower flows in summer and 13% higher flows in early fall.  Overall, Marmot Creek basin annual 30   streamflow discharge will increase by 18% with PGW without a change in its streamflow generation efficiency, despite the basin shifting from snowmelt runoff towards rainfall-dominated runoff generation.



## 1 Introduction

The eastern slopes of the Canadian Rocky Mountains form the headwaters of the Saskatchewan River Basin (SRB) and are a vital source of water supply to downstream users in the three Prairie Provinces of Canada. These mountain headwaters occupy about 12.6% of total drainage area but provide about 87% of total water yield for domestic, agricultural, and industrial users
in the SRB (Redmond, 1964). Climate in this region has been experiencing changes since the last century (Whitfield, 2014; DeBeer et al., 2016). A significant warming of 0.5 °C to 1.5 °C has occurred in western Canada over the period 1900-1998, with the greatest increases in winter daily minimum temperature (Zhang et al., 2000). The regional warming has been exceeded in the eastern slopes of the Canadian Rocky Mountains, with mean temperatures increasing by 2.6 °C and winter minimum temperatures increasing by 3.6 °C at middle elevations in Marmot Creek Research Basin (MCRB) since the early 1960s
(Harder et al., 2015). With the warming air temperatures, the rainfall ratio (ratio of rainfall to total precipitation) is increasing as the fraction of precipitation as snowfall declines (Lapp et al., 2005; Shook and Pomeroy, 2012). In mountains of western North America this leads to decreases in the seasonal snowpack (Mote et al., 2005; Brown and Robinson, 2011) and consequently earlier spring runoff (Stewart et al., 2004). In contrast to ubiquitous warming temperatures, trends in precipitation volume are mixed for Canadian Rocky Mountains, as some studies show increasing trends of about 14% in annual precipitation
over the period 1948-2012 (Vincent et al., 2015) and other studies do not find trends or change (Valeo et al., 2007; Harder et al., 2015). With the anticipated changes in the future climate for this region (IPCC, 2013), understanding the impacts of projected climate change on the hydrological cycle in headwater basins is important for future water management in the SRB.

Winter snow accumulation provides the greatest source of streamflow runoff in many mountain regions of world (Grant and Kahan, 1974; Serreze et al., 1999), as snowmelt is the most important annual hydrological event (Gray and Male,
1981). Melt from the seasonal snowpack is the main contributor of streamflow in the eastern slopes of the Canadian Rocky Mountains (Kienzle et al., 2012; Pomeroy et al., 2012; Fang et al., 2013). Streamflow generation in mountain regions is highly variable and is controlled by many biophysical and hydrometeorological factors (Hunsaker et al., 2012; Zhang and Wei, 2014). Elevation affects both air temperature and precipitation; two of most important drivers of snowpack variability in mountains (Lundquist and Cayan, 2007; Marks et al., 2013). Topographic features, such as slope/aspect and forest structure are other
important factors contributing to the heterogeneity of radiation, atmospheric energy and wind flow in mountain environments (Föhn and Meister, 1983; Bernhardt et al., 2009; Marsh et al., 2012; Musselman and Pomeroy, 2017; MacDonald et al., 2018), and result in the high spatial variability of snow accumulation, melt patterns, evapotranspiration, and runoff in complex mountain terrain (MacDonald et al., 2010; Ellis et al., 2013; Revuelto et al., 2014; Knowles et al., 2015; DeBeer and Pomeroy, 2017).

Many studies have examined the impacts of climate change on snow accumulation, redistribution, snowmelt, evapotranspiration, soil moisture storage, and streamflow in alpine watersheds through the simulations of hydrological models driven by future climate scenarios generated by downscaling of climate model outputs or perturbations of current meteorological observations (Kienzle et al., 2012; López-Moreno et al., 2014; Rasouli et al., 2015; Jepsen et al., 2016; Weber





et al., 2016; Meißl et al., 2017). Physically based hydrological models are effective ways to analyse the hydrological response to climate change, as they can capture the complex hydrological processes governing streamflow generation for mountain watersheds. Empirical snow modelling methods that use temperature-index techniques have great difficulty in cold mountain regions (Swanson, 1998) and generally do not perform well because of their lack of physical basis, need for calibration from

sparse snowmelt observations, and neglect of sublimation contributions to ablation (Walter et al., 2005; Pomeroy et al., 2005; 2013). The Cold Regions Hydrological Modelling platform (CRHM; Pomeroy et al., 2007; 2016) offers a full suite of streamflow generation processes for Canadian Rocky Mountains (i.e. wind redistribution of alpine snow, snow avalanching on steep alpine slopes, snow interception, sublimation, drip and unloading from forest canopies, infiltration to frozen and unfrozen soils, overland and detention flow, hillslope sub-surface water redistribution, and evapotranspiration from forests,

clearings and alpine tundra). Physically based algorithms in CRHM have been developed from field studies (Pomeroy et al., 2009; DeBeer and Pomeroy, 2010; Ellis et al., 2010; MacDonald et al., 2010; Harder and Pomeroy, 2013) and have been extensively evaluated in mountain headwater basins where models created using CRHM can be run successfully without calibration from streamflow (Fang et al., 2013; Pomeroy et al., 2013; Rasouli et al., 2015; Fang and Pomeroy, 2016; Pomeroy et al., 2016).

15         A recent application of the Weather Research and Forecasting (WRF) model provides 4 km simulation outputs for both current climate and a future climate scenario using dynamical downscaling from reanalysis data for large portions of North America with perturbations from an ensemble of Regional Climate Model (RCM) projections called pseudo global warming (PGW) as discussed by Liu et al. (2017) and Li et al. (2019). This remarkably high resolution WRF application permits convective precipitation processes, resolves mountain topography and so can capture variations in surface meteorology

due to mesoscale orography such as found in mountains. The objectives of this paper are to combine the climate predictions from WRF with the hydrological predictions from CRHM to: (1) evaluate the ability to simulate snowpack and streamflow regimes in a Canadian Rocky Mountain headwater basin without calibration, using a dynamically downscaled atmospheric model coupled to a physically based cold regions hydrology model; (2) diagnose the detailed changes in hydrology due to impending climate change for this headwater basin using cold regions hydrology simulations driven by dynamically

downscaled current and future climate outputs.

## 2 Methods

### 2.1 Study site

The study was conducted in the Marmot Creek Research Basin (MCRB) (50.95°N, 115.15°W) in the Kananaskis Valley, Alberta, Canada, located in eastern slopes of the Canadian Rockies (Fig. 1). MCRB is a small headwater basin (9.4 km$^2$) of

the Bow River Basin and flows into the Kananaskis River. MCRB is composed of three upper sub-basins: Cabin Creek (2.35 km$^2$), Middle Creek (2.94 km$^2$), and Twin Creek (2.79 km$^2$), which converge into the confluence sub-basin above the main stream gauge (1.32 km$^2$). The basin elevation ranges from 1590 m a.s.l. (above sea level) at the main Marmot Creek outlet to



2829 m at the summit of Mount Allan. Most of MCRB is covered by needleleaf forest; Engelmann spruce (*Picea engelmanni*) and subalpine fir (*Abies lasiocarpa*) are dominant in the upper part of basin (1710 to 2277 m). The lower elevation (1590 to 2015 m) forests are mainly Engelmann spruce and lodgepole pine (*Pinus contorta* var. Latifolia) with aspen woodland near the basin outlet (Kirby and Ogilvy, 1969). Alpine larch (*Larix lyallii*) and short tundra shrubs are found around the treeline at

approximately 2016 to 2379 m. Exposed rock surfaces and talus predominate in the high alpine part of basin (1956 to 2829 m). Forest management experiments conducted in the 1970s and 1980s left large clearcut blocks (1838 to 2062 m) in the Cabin Creek sub-basin and numerous small circular clearings (1762 to 2209 m) in the Twin Creek sub-basin (Golding and Swanson, 1986). Physiographic descriptions of these ecozones are shown in Table 1. The surficial soils are poorly developed mountain soils consisting principally of glaciofluvial, surficial till and postglacial colluvium deposits (Beke, 1969). Relatively

impermeable bedrock is found near the surface at the higher elevations and headwater areas, whilst the rest of basin is covered by a deep layer of coarse and permeable soil allowing for rapid rainfall infiltration to subsurface layers overlying relatively impermeable shale (Jeffrey, 1965). Continental air masses control the weather in the region, which has long and cold winters and cool and wet springs with a late spring/early summer precipitation maximum that can fall as rainfall or snowfall. Westerly warm and dry Chinook (foehn) winds lead to brief periods with air temperatures well above 0 °C during the winter months.

Annual precipitation ranges from 600 mm at lower elevations to more than 1100 mm at the higher elevations, of which approximately 70 to 75% occurs as snowfall with the percentage increasing with elevation (Storr, 1967). Mean monthly air temperatures range from 14 °C observed at 1850 m in July to -10 °C observed at 2450 m in January.

## 2.2 WRF model

### 2.2.1 Model overview

The Weather Research and Forecasting (WRF) model Version 3.4.1 is used in this paper. This version of WRF permits convective weather system at sub-synoptic scales and deals with mesoscale orography at a 4 km horizontal grid spacing for large portions of North America. Two 13-year experiments were conducted, consisting of a control (CTRL) simulation and a pseudo global warming (PGW) simulation. The CTRL simulation is retrospective for 2000-2013 period with initial and boundary conditions from 6-hour 0.703° ERA-Interim reanalysis data (Dee et al., 2011). The PGW simulation is a 13-year

(i.e. 2000-2013) simulation forced with the 6-hour ERA-Interim reanalysis data plus a climate perturbation. The climate perturbation was derived from 19-model ensemble mean change from the fifth phase of the Coupled Model Intercomparison Project (CMIP5; Taylor et al., 2012) under a "business as usual" forcing scenario: representative concentration pathway 8.5 (RCP8.5; van Vuuren et al., 2011). The PGW simulation is equivalent to a future climate scenario for the 2086-2099 period. The perturbation for the WRF PGW is different from monthly perturbed climate that is based on differences in monthly 30-

year means between current and future climate in 11 RCM outputs described in Rasouli et al. (2019). A detailed description of the WRF model setup is provided by Li et al. (2019). For MCRB, the 4 km hourly WRF outputs from both CTRL and PGW simulations were extracted for the WRF gird shown in Fig. 1. The extracted variables include near-surface air temperature,





vapour pressure, wind speed, precipitation, and shortwave irradiance; relative humidity is required by CRHM and was estimated with conversion equations using air temperature and vapour pressure (Tetens, 1930).

### 2.2.2 Bias correction

Although the 4 km WRF model allows direct use of microphysics and resolves mesoscale convection, which provides a
considerable level of spatial detail, it still produces biases in the near-surface meteorology for MCRB. The bias is caused by complex mountain terrain in MCRB, and many of the topographic features such as alpine ridges, wind exposed and wind sheltered slopes, and valley bottoms influence the distribution of near-surface meteorological conditions at scales smaller than 4 km (Vionnet et al., 2015). Thus, the extracted 4 km WRF outputs were bias corrected using the quantile delta mapping (QDM) algorithm (Cannon et al., 2015). The QDM algorithm corrects systematic bias in quantiles of WRF outputs with
respect to the observations and preserves model projected relative changes in quantiles. QDM first extracts the climate change signal from projected future quantiles and then detrends the series before reintroducing the trends in projected future quantiles. A transfer function that transforms the cumulative distributions of the model outputs to match the cumulative distributions of observed data is used in QDM to correct both historical and projected model outputs. More details of QDM are provided by Cannon et al. (2015). Observations of air temperature, relative humidity, wind speed, precipitation, and shortwave irradiance
from the Centennial Ridge, Fisera Ridge, Vista View, Upper Clearing and Upper Forest hydrometeorological stations were used in the QDM algorithm. These stations are shown in Fig. 1 and are described in several publications (DeBeer and Pomeroy, 2010; Ellis et al., 2010; MacDonald et al., 2010; Pomeroy et al., 2012). This bias correction downscaled the WRF grid with respect to these stations and created sub-grid WRF surface meteorology at these stations that are treated as virtual stations used to force hydrological simulations. For the WRF CTRL outputs, the bias correction was performed for eight water years (i.e. 1
October to 30 September) from 2005 to 2013; this is the overlapping period for WRF CTRL outputs and observations in MCRB. The WRF PGW outputs were bias corrected by preserving the model projected relative changes in quantiles, resulting in eight water years of corrected WRF PGW outputs from 1 October 2091 to 30 September 2099. Statistical indexes used to assess WRF CTRL outputs were the root mean square difference (RMSD) calculated as in Fang et al. (2013) and the mean absolute difference (MAD) computed as follows:

$$MAD = \frac{1}{n}\sum |x_s - x_o|, \tag{1}$$

where $n$ is number of samples, and $x_s$ and $x_o$ are the observed and modelled values, respectively.

### 2.3 Hydrological model and simulations

The Cold Regions Hydrological Modelling platform (CRHM) was used to create a hydrological model for MCRB. CRHM is
an object-oriented, modular and flexible platform for assembling physically based hydrological models. With CRHM, the user constructs a purpose-built model from a selection of possible basin spatial configurations, spatial resolutions and physical





process modules of varying degrees of physical complexity. Basin discretization is performed via dynamic networks of hydrological response units (HRUs) whose number and nature are selected based on the variability of basin attributes and the level of physical complexity chosen for the model. Physical complexity is selected by the user in light of hydrological understanding, parameter availability, basin complexity, meteorological data availability and the objective flux or state for

prediction. A full description of CRHM is provided by Pomeroy et al. (2007). For MCRB, a set of physically based modules was assembled to simulate the dominant hydrological processes by Pomeroy et al. (2012) and Fang et al. (2013), including wind redistribution of alpine snow, snow interception, sublimation, drip and unloading from forest canopies, sub-canopy radiation energetics, slope/aspect effects on radiation and wind flow, infiltration to frozen and unfrozen soils, overland flow, hillslope sub-surface water flow and storage, and evapotranspiration from forests, clearings and alpine tundra. Recent updates

were made to the evaporation and hillslope modules for better representation of runoff on hillslopes and evapotranspiration from vegetation with seasonal variations in leaf area index and height, and the updated model was evaluated in the June 2013 flood (Fang and Pomeroy, 2016; Pomeroy et al., 2016).

Hydrological model simulations were conducted with the bias-corrected WRF near-surface meteorological variables: air temperature, relative humidity, wind speed, precipitation, and shortwave irradiance for CTRL and PGW periods,

respectively. Simulations in both periods cover eight water years: CTRL for current period (i.e. 1 October 2005 to 30 September 2013) and PGW for future period (i.e. 1 October 2091 to 30 September 2099). Model simulations of snow accumulation, spring snowmelt, and streamflow in CTRL period were evaluated against the observations of snow accumulation, snowmelt, and streamflow in MCRB. Statistical indexes used to evaluate the model simulations in CTRL period were Nash Sutcliffe efficiency (NSE) (Nash and Sutcliffe, 1970) and other indexes – root mean square difference (RMSD),

normalised RMSD (NRMSD), and model bias (MB) calculated by equations shown in Fang et al. (2013). Then, water balance variables and radiation fluxes for all ecozones in MCRB shown in Table 1 and basin streamflow from model simulations in CTRL and PGW periods were compared and used to diagnose the changes in hydrology for all ecozones and for the whole MCRB.

## 3 Results

### 3.1 WRF CTRL outputs

Near-surface hourly air temperature, relative humidity, wind speed, precipitation, and shortwave irradiance from observations, uncorrected WRF CTRL outputs, and bias corrected WRF CTRL outputs were compared for the Centennial Ridge, Fisera Ridge, Vista View, Upper Clearing and Upper Forest stations in MCRB. Figure 2 shows the quantile-quantile (Q-Q) plots of air temperature, relative humidity, wind speed, precipitation, and shortwave irradiance for the well-exposed Fisera Ridge

station from observations, WRF CTRL outputs, and bias corrected WRF CTRL outputs. The points in the Q-Q plots of observations and bias corrected WRF outputs shown in Fig. 2b, 2d, 2f, 2h, and 2j are linearly distributed on the 1:1 line, while the points in the Q-Q plots of observations and uncorrected WRF outputs shown in Fig. 2a, 2c, 2e, 2g, and 2i do not appear to



be linear distribution. This suggests that the near-surface meteorological variables from observations and bias corrected WRF CTRL outputs form the same distribution. The Q-Q plots for other stations show the same results and are provided as Supplement. Table 2 shows MAD and RMSD indexes for accessing the WRF CTRL outputs. Values of the MAD for the bias-corrected WRF outputs were zero and were smaller than those for the uncorrected WRF outputs, suggesting there is no

difference in the statistical distributions of observations and bias corrected WRF CTRL outputs. Values of the RMSD for the bias-corrected WRF outputs were lower than those for the uncorrected WRF outputs, except for wind speed for Centennial Ridge station and precipitation for Fisera Ridge station. Values of RMSD were 4.88 m s$^{-1}$ and 0.6 mm for bias-corrected WRF wind speeds for the Centennial Ridge station and bias-corrected WRF precipitation for the Fisera Ridge station, respectively, and were slightly higher than the RMSD of 4.61 m s$^{-1}$ and 0.56 mm for the original WRF wind speed and precipitation. Despite

that, the bias correction generally improved WRF outputs and reduced the mean difference compared to the observations.

**3.2 Hydrological model evaluations**

CRHM simulations of the snow water equivalent accumulation (SWE) using bias-corrected WRF CTRL near-surface outputs were compared to observed SWE for the sheltered, mid-elevation Upper Forest and Upper Clearing sites (Fig. 3a-b) and for the wind-blown, high-elevation Fisera Ridge site (Fig. 3c-f) for 2007-2013. The results demonstrate that the model forced

with the bias-corrected WRF outputs was able to simulate SWE for both forest and alpine environments, with exceptions for the Upper Forest site during season of 2007/2008. In addition, Table 3 shows that the simulations had large differences with the observations for both forest and alpine sites during the season of 2011/2012, with RMSD ranging from 52.3 to 297.5 mm for spruce forest and lower alpine south-facing slope and MB ranging from -0.65 to -0.35 for spruce forest and larch forest treeline, respectively. For all six seasons, model simulations captured the general seasonal patterns of snow accumulation and

ablation for these forest and alpine sites, with MB values of all seasons ranging from -0.43 for the forest clearing to 0.17 for the alpine ridge top (Table 3). All season RMSD ranged from 46.5 mm for the mature spruce forest to 260.1 mm for the larch forest treeline, while the NRMSD ranged from 0.39 for upper alpine south slope to 0.84 for the mature spruce forest.

Further model evaluation was conducted using the CRHM simulated streamflow driven by bias-corrected WRF CTL outputs and the observed outlet streamflow discharge measured by the Water Survey of Canada (WSC) gauge (05BF016) for

2005-2013 (Fig. 4). The WSC gauged streamflow from 1 May to 30 September during 2006-2012 and for part of 2013 before the flood. Streamflow during 2013 flood was estimated by the University of Saskatchewan Centre for Hydrology (CH) using the best available information but with great uncertainty (Harder et al., 2015). After the flood, a streamflow gauge established near the same outlet by CH continued measurements for the remainder of 2013. The seasonal NSE values ranged from -0.33 in 2009 to 0.72 in 2012, and the overall eight-season NSE was 0.4 (Table 4), suggesting model had some predictability for the

temporal evolution of daily basin discharge in these eight seasons. The eight-season RMSD, NRMSD and MB listed in Table 4 were 0.212 m$^3$ s$^{-1}$, 0.79 and 0.001 for the predicted daily basin discharge, respectively, indicating relatively small differences between the simulated and observed Marmot Creek streamflow.



### 3.3 Changes in WRF meteorology due to climate change

The WRF near-surface meteorological variables that had been bias-corrected with respect to MCRB stations were compared for eight water years (WY) between CTRL (i.e. 1 October 2005 to 30 September 2013) and PGW (i.e. 1 October 2091 to 30 September 2099). Figure 5 shows the comparisons of mean WY air temperature, relative humidity, wind speed and shortwave
irradiance and cumulative WY precipitation at the Centennial Ridge, Fisera Ridge, Vista View, Upper Clearing and Upper Forest stations. There were consistent increases in mean WY air temperatures in PGW for all MCRB stations compared to those in CTRL. Compared to the eight-WY mean temperatures in CTRL which ranged from -1.8 °C at Centennial Ridge to 1.4 °C at Vista View, those in PGW ranged from 2.9 °C at Centennial Ridge to 6.1 °C at Vista View and so were about 4.7 °C warmer for all stations (Fig. 5a). The mean WY relative humidity decreased slightly in PGW compared to that in CTRL, with
the eight-WY mean humidity declining by 1.8% for all stations (Fig. 5b), whilst the mean WY wind speed remained unchanged between CTRL and PGW (Fig. 5c). There were slight increases in the mean WY shortwave irradiance in PGW compared to CTRL, and the eight-WY mean shortwave irradiance increased by 2.8 W m$^{-2}$ at both Centennial Ridge (i.e. from 141.4 to 144.2 W m$^{-2}$) and Fisera Ridge (i.e. from 152.4 to 155.2 W m$^{-2}$) and increased by 2.1 W m$^{-2}$ at Upper Clearing (i.e. from 140.7 to 142.8 W m$^{-2}$) (Fig. 5d). There was much more precipitation in PGW, and annual precipitation  was 1287 and 882 mm over
eight-WY in PGW at Fisera Ridge and Upper Clearing, respectively, about 147 and 150 mm more or 13% and 20% increases compared to those in CTRL at Fisera Ridge and Upper Clearing (Fig. 5e).

### 3.4 Changes in water balance variables

The simulated annual water balance variables for all ecozones were compared for eight WY between CTRL (i.e. 1 October 2005 to 30 September 2013) and PGW (i.e. 1 October 2091 to 30 September 2099). Compared to CTRL, rainfall increased
and snowfall decreased in PGW for all ecozones (Fig. 6a-b). The increase in the annual rainfall ranged from 201 mm at lower forest to 328 mm at alpine, whilst the decrease in the annual snowfall ranged from 63 mm at lower forest to 168 mm at alpine. On average for the whole basin, the eight-WY annual rainfall rose by 268 mm and that of snowfall declined by 112, and there was a 156 mm or 16% increase in total precipitation in PGW (Table 5). Actual evapotranspiration is the sum of evaporation from soil, forest canopy rain interception and open water, and transpiration from plants; this increased in PGW for all ecozones
because of more rainfall (Fig. 6c). The increase in annual actual evapotranspiration over eight WY ranged from 59 mm at alpine to 179 mm at upper forest, with 124 mm increase for the whole basin shown in Table 5. Sublimation is the total of blowing snow, surface snowpack and forest canopy interception sublimation and declined in PGW for all ecozones as a result of decreased snowfall and the impact of warmer air temperatures in limiting blowing snow occurrence and increasing unloading of intercepted snow from forest canopies; the decrease in annual sublimation ranged from 6 mm at the forest circular clearing
north-facing to 58 mm at the alpine (Fig. 6d), with a reduction of 40 mm for entire basin (Table 5). For the sheltered and sparsely vegetated forest clearing ecozones, neither blowing snow nor forest interception processes occur, so there was only change in surface snowpack sublimation for these ecozones. For the alpine and treeline ecozones, blowing snow is a very





important process in controlling seasonal snow accumulation; the alpine is a source of blowing snow (i.e. negative value for blowing snow transport) whilst the treeline receives blowing snow which accumulates in deep snowdrifts (i.e. positive value for blowing snow transport). Fig. 6e shows that blowing snow transport was suppressed in PGW, resulting in a lower annual blowing snow transport loss from the alpine of 24 mm and a lower annual blowing snow transport gain to treeline of 53 mm.

Blowing snow does not occur in ecozones below the treeline, where no change occurred in blowing snow transport. Annual surface runoff increased by 250 mm in PGW for the alpine ecozone (Fig. 6f), with 87 mm more for entire basin (Table 5). In other, non-alpine, ecozones, flow predominantly occurred in the subsurface and did not show change in the surface runoff between CTRL and PGW. Annual subsurface flow from alpine and treeline ecozones decreased with PGW by 61 and 24 mm, respectively. Although there were increases in annual subsurface flow in PGW for other ecozones, ranging from 8 mm at

lower forest to 56 mm at forest circular clearing north-facing (Fig. 6g), on average for the whole basin, there was 12 mm reduction in annual subsurface flow in PGW (Table 5). Groundwater flow stayed relatively constant between CTRL and PGW for all ecozones (Fig. 6h). There were increases in annual total storage in soil and groundwater in PGW for some ecozones, ranging from 9 mm at upper forest to 27 mm at forest circular clearing south-facing, whilst the total subsurface storage dropped by from 2 mm at lower forest to 40 mm at forest clearing blocks ecozone under PGW (Fig. 6i). Annual subsurface storage

declined by 12 mm, from 416 mm in CTRL (i.e. 45% saturation) to 404 mm in PGW (i.e. 43% saturation) for the entire basin (Table 5).

**3.5 Changes in snow regime**

The simulated snow accumulation (SWE) for all ecozones and the entire basin was compared for eight WY between CTRL (i.e. 1 October 2005 to 30 September 2013) and PGW (i.e. 1 October 2091 to 30 September 2099). Figure 7 illustrates the

annual time-series of SWE for CTRL and PGW and demonstrates the impacts on seasonal SWE by PGW climate for different ecozones. For all ecozones, the peak SWE occurred earlier and was much lower in PGW compared to CTRL, with decreases ranging from 7 mm at upper forest to 166 mm at treeline. In the alpine ecozone, snowpack in the cold mid-winter was not impacted by PGW climate and had very comparable and even higher value before early April compared to that in CTRL, but after that SWE ablated rapidly and disappeared earlier - by about 26 days, in PGW. For other ecozones, the seasonal snowpack

underwent substantial declines and decreased throughout the season in PGW, and the date of seasonal snowpack depletion advanced from early August to late June at treeline and from mid-June to late May in the other ecozones. For eight WY, the mean melt rate was estimated by dividing mean annual peak SWE by the number of days from peak SWE to snowpack depletion and was lower for treeline and forest clearings ecozones in PGW, with decreases ranging from 0.9 mm day$^{-1}$ at forest clearing blocks (i.e. from 1.4 mm day$^{-1}$ in CTRL to 0.5 mm day$^{-1}$ in PGW) to 1.6 mm day$^{-1}$ at forest circular clearing north-

facing (i.e. from 2.9 mm day$^{-1}$ in CTRL to 1.3 mm day$^{-1}$ in PGW). Whilst the melt rate was slightly higher for alpine and forests ecozones, with increases ranging from 0.01 mm day$^{-1}$ at upper forest (i.e. from 0.63 mm day$^{-1}$ in CTRL to 0.64 mm day$^{-1}$ in PGW) to 0.04 mm day$^{-1}$ at both alpine (i.e. from 1.95 mm day$^{-1}$ in CTRL to 1.99 mm day$^{-1}$ in PGW) and lower forest





(i.e. from 0.46 mm day$^{-1}$ in CTRL to 0.50 mm day$^{-1}$ in PGW). For the entire basin, there was a very small decline in the melt rate from 1.3 mm day$^{-1}$ in CTRL to 1.1 mm day$^{-1}$ in PGW (Table 5).

Changes in the seasonal total snowmelt, peak SWE, snowcover duration and radiation fluxes to snowcover from eight WY were also compared between CTRL and PGW. Figure 8a shows that cumulative snowmelt volume decreased in PGW

for all ecozones, and for the eight-WY mean total snowmelt, treeline suffered highest decrease by 215 mm, with the declines ranging from 32 mm at upper forest to 113 mm at alpine. The peak SWE of seasonal snowpack reduced for all ecozones in PGW, and decrease in the mean value of eight-WY peak SWE was highest at treeline by 149 mm and lowest at upper and lower forests by 11 mm (Fig. 8b). The duration of seasonal snowpack became shorter for all ecozones in PGW, with the eight-WY mean snowcover duration shortened by 31 days at forest circular clearing north-facing to 49 days at treeline (Fig. 8c).

Table 5 shows that for basin, eight-WY mean snowmelt volume, peak SWE and duration of seasonal snowpack decreased by 84 mm, 40 mm and 49 days, respectively. The annual net radiation to snowcover increased in PGW for all ecozones, ranging from 2 W m$^{-2}$ at lower forest to 4 W m$^{-2}$ at other ecozones (Fig. 8f). The increases in the net radiation to snowcover was because of higher annual longwave irradiance to snowcover in PGW for all ecozones, ranging from 10 W m$^{-2}$ higher at lower forest to 17 W m$^{-2}$ higher at treeline and forest clearings ecozones (Fig. 8e), whilst the annual solar irradiance to snowcover

reduced for all ecozones, with declines ranging from 2 W m$^{-2}$ at upper forest to 17 W m$^{-2}$ at forest clearing blocks (Fig. 8d). For the entire basin in PGW, annual solar irradiance and longwave to snowcover decreased by 11 W m$^{-2}$ and increased by 14 W m$^{-2}$, respectively, with 4 W m$^{-2}$ increase in annual net radiation to snowcover (Table 5).

### 3.6 Changes in streamflow

Simulated daily streamflow discharge was compared for the eight WY between CTRL (i.e. 1 October 2005 to 30 September

2013) and PGW (i.e. 1 October 2091 to 30 September 2099). Figure 9a shows the annual time-series of Marmot Creek basin discharge for CTRL and PGW and illustrates that the basin discharge was very similar between CTRL and PGW before mid-March, suggesting the discharge was not greatly impacted by PGW climate in the winter months. However, the average onset of spring freshet advanced by 45 days from 8 May for CTRL to 24 March for PGW, with the centre of flow volume occurring 12 days earlier from 22 June for CTRL to 10 June for PGW. The peak basin discharge was 1.13 m$^3$ s$^{-1}$ and 1.01 m$^3$ s$^{-1}$ in

CTRL and PGW, respectively, both on 21 June. Compared to CTRL, the daily discharge declined in the recession limb between the peak discharge date and late August for PGW period because of higher evaporation in PGW shown in Fig. 6c. The cumulative discharge volume increased by 18% from 3973 dam$^3$ in CTRL to 4683 dam$^3$ in PGW (Fig. 9b). In addition, simulated monthly streamflow discharge was compared for the eight WY between CTRL and PGW for March to October, and change in the monthly discharge was calculated by subtracting monthly discharge in PGW by that in CTRL. Figure 10 shows

noticeable increases in monthly discharge in PGW for months of March to May and September, with increase ranging from 0.02 m$^3$ s$^{-1}$ in March to 0.27 m$^3$ s$^{-1}$ in May compared to monthly discharge in CTRL of 0.01 m$^3$ s$^{-1}$ in March and 0.13 m$^3$ s$^{-1}$ in May. Whilst monthly discharge in PGW declined notably in June and July by 0.03 m$^3$ s$^{-1}$ and 0.096 m$^3$ s$^{-1}$ from monthly discharge in CTRL of 0.69 m$^3$ s$^{-1}$ in June and 0.29 m$^3$ s$^{-1}$ in July, respectively. Monthly discharge in PGW decreased by only





0.01 m³ s⁻¹ in August compared to that of 0.13 m³ s⁻¹ in CTRL and had very similar low value of 0.06 m³ s⁻¹ to that in CTRL in October. For percentage change, monthly discharge in PGW ranged from a 573% increase in April to a 33% decrease in July, and on the seasonal basis, streamflow was 236% higher in spring months (i.e. March to May), 12% lower in summer (i.e. June to August) and 13% higher in early fall (i.e. September to October).

5    The simulated daily runoff fluxes (surface, sub-surface and groundwater runoff) and annual runoff volumes were also plotted for all ecozones in Marmot Creek to examine their changes between CTRL and PGW. Figures 11-12 consistently show minimal change in winter months and an advance in the onset of spring freshet for all ecozones with PGW. The annual peak runoff from the alpine decreased from 25.6 mm day⁻¹ in CTRL to 23.2 mm day⁻¹ in PGW, both occurring on 20 June (Fig. 11a). While the largest decline in annual peak runoff occurred from the treeline, from 27.8 mm day⁻¹ in CTRL to 19.5 mm day⁻¹ in PGW, on 21 June and 17 June, respectively (Fig. 11b). There were moderate declines in annual peak runoff in PGW from other ecozones, ranging 0.3 mm day⁻¹ at lower forest to 2.0 mm day⁻¹ at forest circular clearing north-facing, with no change to the date of annual peak runoff at forest clearing north-facing to that occurring 9 days later at forest clearing south-facing (Figs. 11c-g). There were moderate increases in the annual runoff volume from forest clearing and lower forest ecozones in PGW, ranging from 12 dam³ increase at lower forest to 17 dam³ increase at forest clearing blocks (Figs. 12d-g). The annual runoff volume from the upper forest increased from 258 dam³ in CTRL to 316 dam³ in PGW (Fig. 12c). For alpine and treeline ecozones, primary sources for Marmot Creek basin discharge, the annual runoff volume increased substantially from the alpine from 2457 dam³ in CTRL to 3065 dam³ in PGW (i.e. about 25% increase) but decreased from the treeline from 1007 dam³ in CTRL to 986 dam³ in PGW (i.e. about 2% decline) (Figs. 12a-b).

The relationship between rainfall ratio (RR) and runoff efficiency (RE) were examined for all ecozones and entire basin in CTRL and PGW. A rainfall ratio is defined as total rainfall divided by total precipitation for a water year, and a runoff efficiency is defined as total runoff (surface, subsurface and groundwater runoff) divided by total precipitation for a water year. A RR > 0.5 indicates a rainfall-dominated precipitation regime, and a RR < 0.5 indicates a snowfall-dominated precipitation regime. The RE describes the fraction of precipitation volume that is transformed to runoff by different ecozones in the basin and it normally varies between 1 and 0. Figure 13 illustrates the changes in mean values of RR and RE for all ecozones and the whole basin between CTRL and PGW. The mean RR in CTRL was 0.43 for alpine, meaning it is snowfall-dominated, and that for treeline is 0.51, closed to an equal snowfall and rainfall precipitation regime. For other ecozones, the mean RR were between 0.6 and 0.62 indicating rainfall dominance in CTRL. In contrast, the mean RR increased in PGW and ranged from 0.61 at alpine to 0.76 at lower forest, and for entire basin, the mean RR rose from 0.52 in CTRL (i.e. closed to equal snowfall and rainfall) to 0.68 in PGW (i.e. rainfall-dominated). The mean RE stayed relatively unchanged for the forest ecozones, ranging from 0.1 for lower forest to 0.11 for upper forest in both CTRL and PGW. For the forest clearing ecozones, mean RE values dropped by 0.02. The mean RE had large changes in alpine and treeline ecozones; it dropped from 1.04 in CTRL to 0.91 in PGW for treeline but increased from 0.62 in CTRL to 0.69 in PGW for alpine. For the entire basin, the mean SGE increased by only 0.01 from 0.44 in CTRL to 0.45 in PGW despite basin shifting towards domination by rainfall.



## 4 Discussion

The dynamical downscaling 4 km WRF near-surface meteorology outputs were bias corrected using quantile delta mapping method with respect to station data during October 2005-Septmeber 2013 at MCRB and then were used to force hydrological model simulations in CTRL (i.e. 1 October 2005 to 30 September 2013) and PGW (i.e. 1 October 2091 to 30 September 2099)

periods. The aim is to diagnose the changes in hydrology for this small mountain headwater basin of South Saskatchewan River at end of the 21$^{st}$ century. Results show that the bias corrected WRF CTRL near-surface meteorological variables were comparable to station observations, and the CRHM simulations driven by the bias corrected WRF CTRL near-surface meteorology achieved reasonable predictions for seasonal snow accumulation, melt, and streamflow at MCRB when comparing to the field measurements. WRF simulations can be conducted for higher resolution below 4 km, which provides

more realistic precipitation patterns, especially for extreme events (Tao et al., 2016; Li et al., 2017). However, these higher resolution WRF simulations requires more computations and are still under experiment and evaluation stage, and the 4 km WRF simulations are currently one of the best options to assess future climate change for large region (Liu et al., 2017; Prein et al., 2017).

Results show that the MCRB on average warmed up by 4.7 °C for the high-end scenario (i.e. RCP8.5) in PGW period,

and this warming degree is comparable to other findings for this region by the end of the 21$^{st}$ century (Nogués-Bravo et al., 2007; Kienzle et al., 2012). As air warms, the water holding capacity of atmosphere increases based on the Clausius-Clapeyron equation, resulting in 13 to 20 % more precipitation for different elevations at MCRB in the projected PGW period. The warming future climate directly affects water availability in Rocky Mountain region but sometimes produces complex streamflow responses in this region. Some study suggests declines in the river flows in the 21$^{st}$ century for the central Rocky

Mountain region corresponding to reducing annual precipitation (Rood et al., 2005), while other reported decreases in annual streamflow despite projected increases in precipitation (Tanzeeba and Gan, 2012). These complex streamflow responses are associated with variable influences of surface and subsurface hydrology on streamflow, especially in mountain headwater basins. MCRB is composed of a number of landcovers that span large elevational gradient (i.e. ecozone in this study), and there are different dominant hydrological processes in each ecozone, leading to variable responses to the warming climate.

Results demonstrate that annual precipitation shifted towards more rainfall and less snowfall for all ecozones, with larger changes to alpine and treeline, and MCRB shifted from a closed to equal snowfall and rainfall basin towards rainfall-dominated in PGW period. As result of decreased snowfall, seasonal snowpack collapsed for all ecozones in PGW period, even though both sublimation losses from blowing snow for alpine and from interception for forest also decreased. Treeline ecozone had the highest seasonal snow accumulation in MCRB and suffered the most loss because of combination of reduced snowfall and

supressed blowing snow transport. These results are fairly intuitive and consistent with previous findings in mountain basins (López-Moreno et al., 2013; Pomeroy et al., 2015; Marty et al., 2017). Melt rates of snowcover examined in this study show mixed responses to the projected PGW climate: lower melt rates for treeline and forest clearings ecozones and slightly higher melt rates for alpine and forest ecozones, and a small declined melt rate for the whole basin. These melt rate responses are





perplexing because of complex interaction among changes of air temperature, albedo decay, and radiation fluxes to snowcover in PGW climate. This finding is somewhat different from slower melt rates under warmer climate reported by Musselman et al. (2017). On the other hand, all ecozones at MCRB ubiquitously experienced earlier depletion of seasonal snowpack and shorter snowcover duration in the projected PGW climate; as a result, onset of snowmelt runoff and basin streamflow shifted

toward earlier spring.  Results also show decreases in the basin streamflow during summertime in PGW period because of higher evaporation loss, and consequently water availability from the basin shifted earlier.  These are common findings of streamflow response in the projected future climate for many snow-dominated mountain basins (Stewart et al., 2004; Barnett et al., 2005; Kienzle et al., 2012; Jepsen et al., 2016) and suggest water management for mountain basins would be impacted by that (Rood et al., 2005; Chen et al., 2006; Bongio et al., 2016).

10        It is interesting to note the variability of runoff response from different ecozones at MCRB to the projected PGW climate.  Despite earlier onset of runoff for all ecozones, the annual runoff volume increased very moderately for forest clearings and lower forest ecozones and increased moderately for upper forest ecozone in PGW climate, as increases in precipitation, particularly rainfall, were consumed by increases in evaporation for these ecozones.  Alpine and treeline ecozones compose about 45% of MCRB, and their runoff response to PGW climate impose dominant influence on basin streamflow

change. Results show the opposite response between alpine and treeline.  For alpine, annual runoff volume increased; this is caused by combination of limited available subsurface storage, increase in rainfall that overwhelmed increase in evaporation loss, ultimately resulting in higher flow in alpine primarily from surface runoff despite the slightly reduced subsurface runoff. In contrast, annual runoff volume decreased for treeline, this is caused by the following factors: large reduction in seasonal snow accumulation due to declined snowfall and supressed blowing snow transport, decreased runoff from declined snowmelt,

higher evaporation that exhausted increased rainfall, with more available subsurface storage, subsequently leading to less flow from subsurface runoff.  Decline in runoff from treeline was compensated by increase in runoff from alpine, which contributes to overall increase in annual basin streamflow discharge by 18% for about 16% increase in precipitation in PGW period. Interestingly, MCRB was reported for its hydrological resilience to changes from historic climate, forest disturbance, and extreme events (Harder et al., 2015); it also exhibits somewhat resilience to the projected future climate from the perspective

of basin RE.  That is, despite that basin shifted towards rainfall-dominated, basin RE value remained almost unchanged between current and future climates.

        In this study, assessment of future changes in hydrology at MCRB held landcover and soil properties static between CTRL and PGW periods. Cautions should be used when interpreting this study's results, as landcover and soil properties could also change in the warmer climate.  Modelling studies in southern Rocky Mountain, USA demonstrate future changes in basin

hydrology and water availability by coupling climate change and landcover disturbance and suggest that feedback of landcover disturbance in the warmer climate should be considered when assessing future impacts on mountain basin hydrology (Buma and Livneh, 2015; McDowell et al., 2016; Bennett et al., 2018). However, landcover change such as forest expansion into alpine under warmer climate is uncertain in Canadian Rocky Mountain because of landscape limitation by the geologic and geomorphic processes (Macias-Fauria and Johnson, 2013). For Canadian Rocky Mountain, more collaboration between





hydrology, ecology and soil science is warranted to examine the regional changes in vegetation and soil for the future climate and to come up with more meaningful scenarios of landcover and soil disturbance under changing climate for more comprehensive assessment of future mountain hydrology.

## 5 Conclusions

A physically based hydrological model was set up using the CRHM platform and was forced by bias corrected 4 km WRF near-surface meteorology outputs in CTRL (i.e. 1 October 2005 to 30 September 2013) and PGW (i.e. 1 October 2091 to 30 September 2099) periods to assess the future changes in hydrology at Marmot Creek Research Basin. Model simulations using the bias corrected WRF outputs in CTRL achieved reasonable predictions for snow accumulation and melt for forest and alpine sites compared to the field observations, with MB ranging from -0.43 for forest clearing to 0.17 for alpine ridge top. Model

streamflow simulation was also evaluated against observed daily basin discharge, which shows some predictability for basin discharge, with values of 0.4 and 0.001 for NSE and MB indexes, respectively. Compared to CTRL period, Marmot Creek on average warmed up by 4.7 °C and received 16% more precipitation for the high-end scenario (i.e. RCP8.5) in PGW period; as a result, rainfall ratio rose for Marmot Creek, with 268 mm increase in rainfall and 112 mm decrease in snowfall. However, changes in basin hydrology were more complex than these in precipitation, as basin is composed of seven ecozones spanning

large elevational gradient. On average in PGW period, all ecozones experienced lower seasonal snowpack, with decrease in peak SWE ranging from 11 mm for upper and lower forests to 149 mm for treeline, shorter snowcover duration ranged from 31 days for forest circular clearing north-facing to 49 days for treeline, and total seasonal snowmelt volume decreased from 32 mm for upper forest to 215 mm for treeline. These changes in PGW were result of interaction from several hydrological processes: suppression of blowing snow transport and sublimation for alpine and treeline ecozones and reduced sublimation

from canopy interception for forest ecozones. Snowmelt rates under climate change declined by 1.1mm day$^{-1}$ for treeline and by 0.9 to 1.6 mm day$^{-1}$ for forest clearings and increased by 0.04 mm day$^{-1}$ for alpine and by 0.01 to 0.04 mm day$^{-1}$ for forest ecozones. Additionally, evaporation loss on average increased in PGW from 59 mm for alpine to 179 mm for upper forest, with 124 mm greater evaporation loss for the whole basin. Streamflow responses to the projected PGW climate were even more variable in the different Marmot Creek ecozones. Runoff began earlier for all ecozones in PGW period, forested and

alpine ecozones generated larger annual runoff volume, from 12% to 25%, whereas the treeline ecozone had a small (2%) in annual runoff volume. For the whole basin, the declining runoff at treeline was compensated by higher runoff at other ecozones, resulting in 18% higher annual basin streamflow discharge in PGW period. Given the 16% increase in precipitation, runoff efficiency for Marmot Creek virtually unchanged for the projected PGW climate despite the hydrological shift from snowmelt-runoff towards rainfall-dominated runoff.

*Data availability*. The dataset is available upon request through the Changing Cold Regions Network (CCRN) database (http://www.ccrnetwork.ca/outputs/data/; CCRN, 2018) and the corresponding author of the paper (xing.fang@usask.ca).



*Author contribution.* XF and JP designed the study. XF performed the WRF bias correction, model simulations, and analyses. XF prepared the manuscript with contributions from JP in the manuscript outline and editing, results analysis and discussion.

*Competing interests.* The authors declare that they have no conflict of interest.

*Special issue statement.* This article is part of the special issue "Understanding and predicting Earth system and hydrological change in cold regions". It is not associated with a conference.

*Acknowledgements.* The authors would like to gratefully acknowledge the funding assistance provided from the Alberta Government departments of Environment and Parks, and Agriculture and Forestry, Alberta Innovates, the IP3 Cold Regions Hydrology Network of the Canadian Foundation for Climate and Atmospheric Sciences, and the Changing Cold Regions Network, University of Saskatchewan Global Institute for Water Security, and the Canada Research Chairs programme. Logistical assistance was received from the Biogeoscience Institute, University of Calgary and the Nakiska Ski Area.
Streamflow data from the Water Survey of Canada is gratefully acknowledged.  Field work by many graduate students in and collaborators with the Centre for Hydrology and research officers Michael Solohub, May Guan and Angus Duncan was essential in accurate data collection in adverse conditions. Dr. Kabir Rasouli is acknowledged for providing source to download WRF .nc files used to extract outputs for this study.

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





**Figure 1.** Location and contour map of the Marmot Creek Research Basin (MCRB), showing hydrometeorological stations, hydrometric station, and WRF grid centroid, and ecozones of the MCRB: alpine, treeline, upper forest, forest clearing blocks, forest circular clearings, and lower forest. Note that the size and areas of circular clearings in Twin Creek are not to scale.



**Figure 2.** Quantile-quantile plots of observations and WRF CTRL outputs for Fisera Ridge station in MCRB: (a) WRF CTRL and observed air temperature, (b) corrected WRF CTRL and observed air temperature, (c) WRF CTRL and observed relative humidity (d) corrected WRF CTRL and observed relative humidity, (e) WRF CTRL and observed wind speed (f) corrected WRF CTRL and observed wind speed, (g) WRF CTRL and observed incoming solar radiation (h) corrected WRF CTRL and observed incoming solar radiation, (i) WRF CTRL and observed precipitation (j) corrected WRF CTRL and observed precipitation. Note that best linear fit is straight line connecting the first and third quartiles.





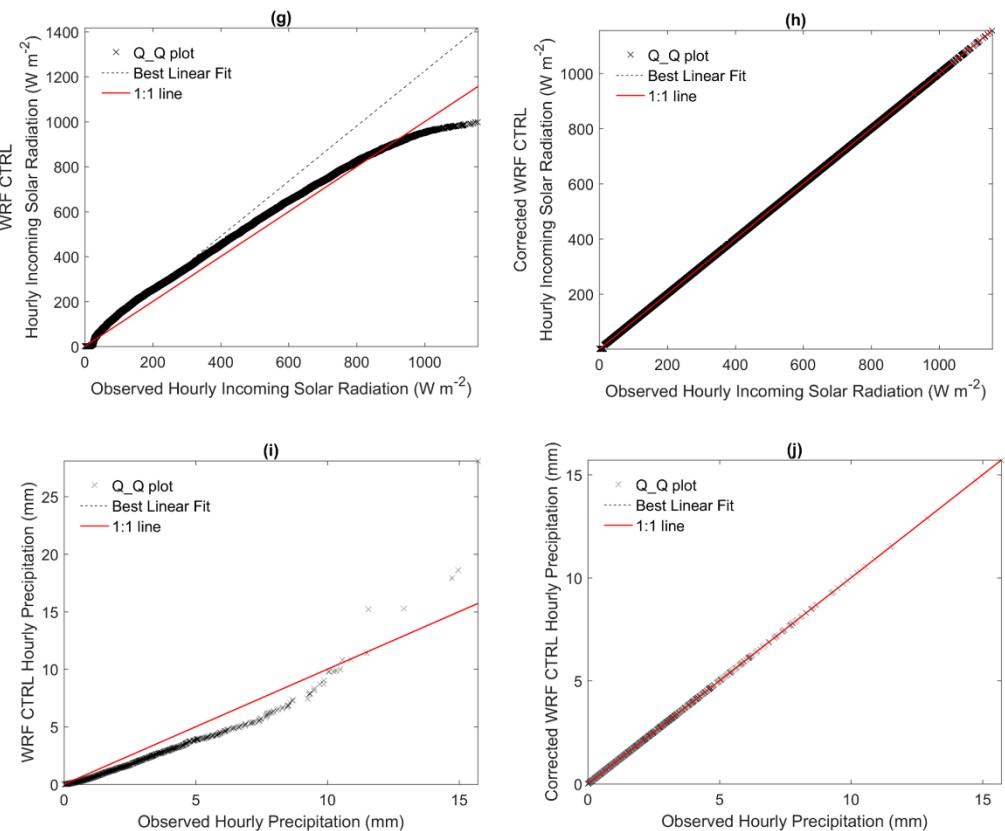

**Figure 2.** Continued.



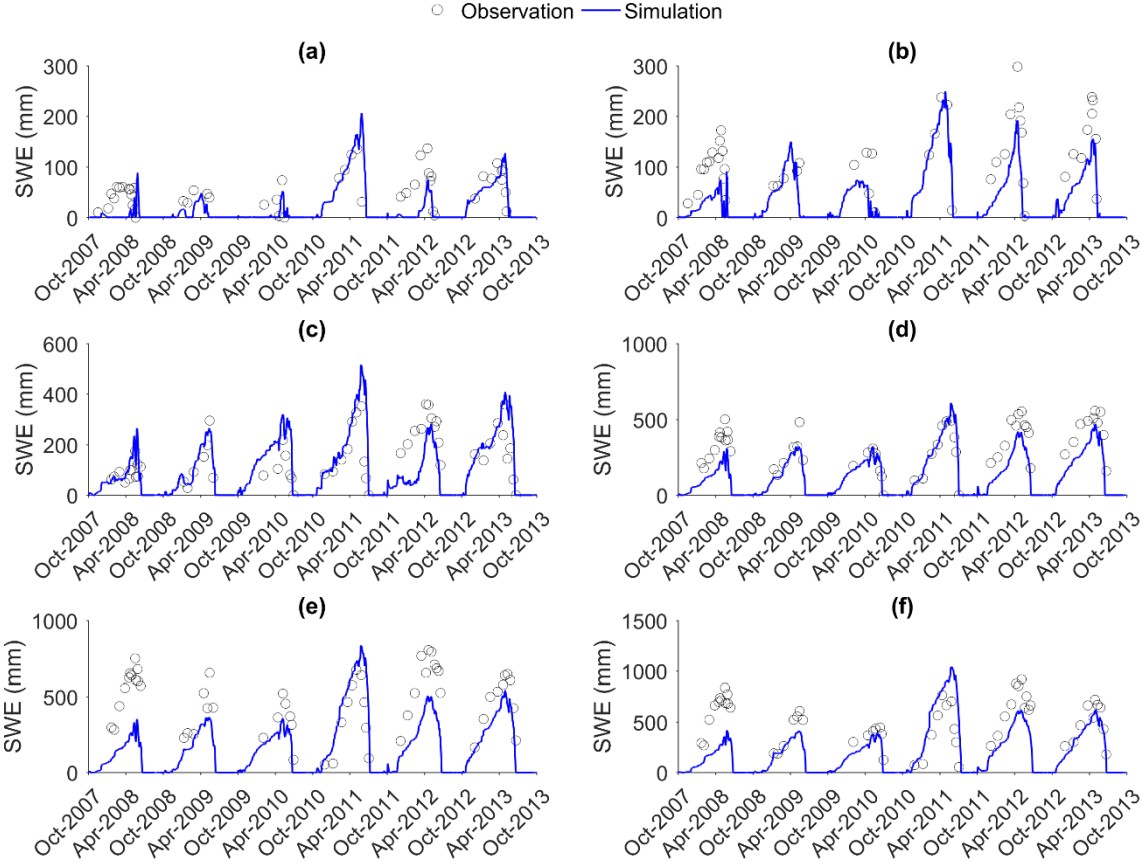

**Figure 3.** Comparisons of the observed and simulated snow accumulation (SWE) for 2007-2013 at the sheltered, mid-elevation Upper Forest/Clearing and the wind-blown, high-elevation Fisera Ridge sites in MCRB. (a) Mature spruce forest, (b) forest clearing, (c) ridge top, (d) upper alpine south-facing slope, (e) lower upper alpine south-facing slope, and (f) larch forest treeline.





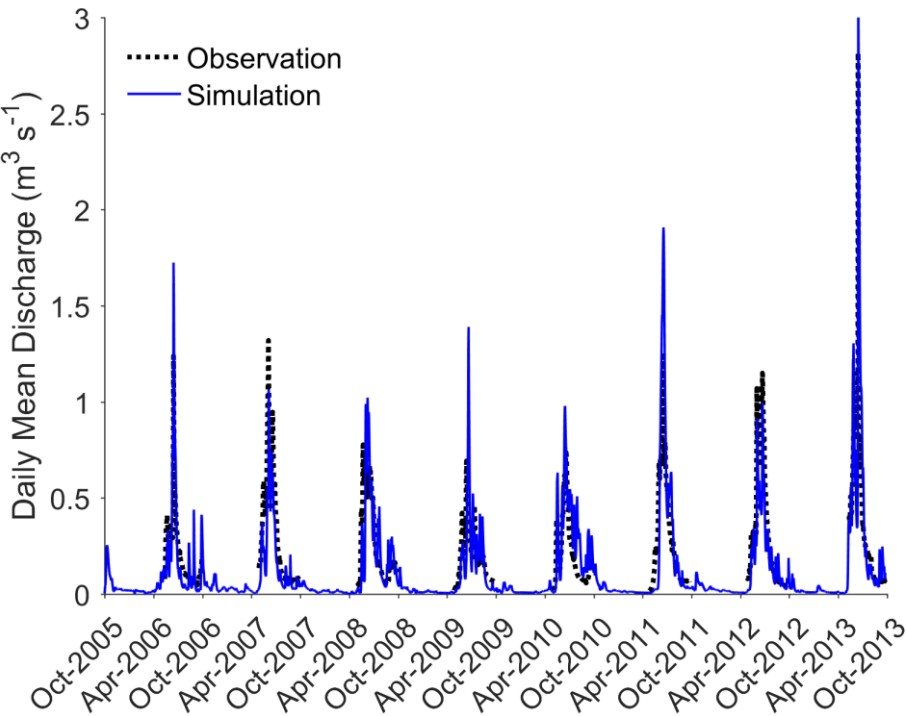

**Figure 4.** Comparisons of the observed and simulated daily streamflow for 2005-2013 for Marmot Creek.



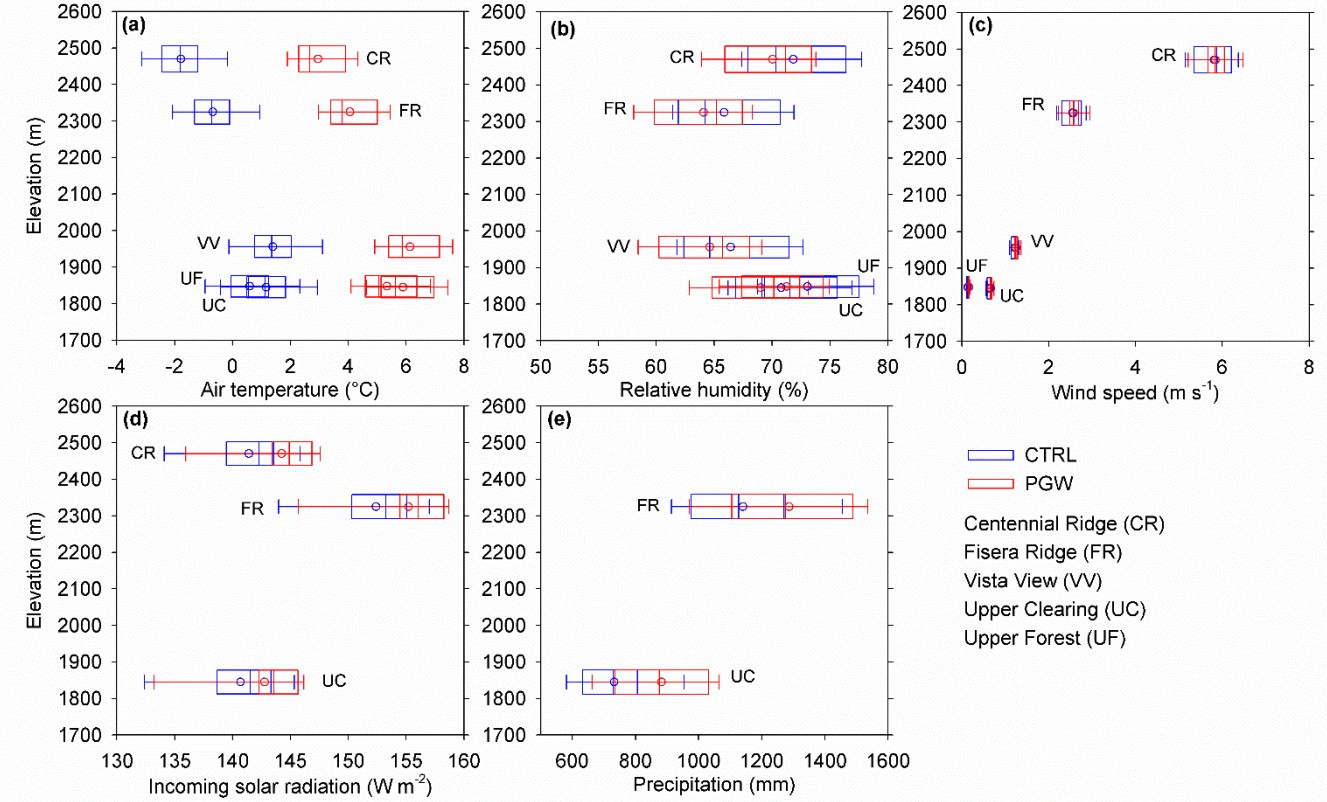

**Figure 5.** Boxplots of the bias corrected WRF CTRL and PGW near-surface meteorology for MCRB station sites. (a) Air temperature, (b) relative humidity, (c) wind speed, (d) incoming solar radiation, and (e) precipitation. Note that total water year precipitation is presented and average water year value is presented for other variables. Vertical line within the box = median value of the eight-water year data, box = interquartile range (Q1: 25% to Q3: 75%) of the eight-water year data, whiskers = minimum and maximum, circle = mean value of the eight-water year data.







**Figure 6.** Boxplots of the simulated annual water balance variables for WRF CTRL and PGW for alpine (AE), treeline (TE), upper forest (UF), forest clearing blocks (FCB), forest circular clearing north-facing (FCCNF), forest circular clearing south-facing (FCCSF), lower forest (LF) ecozones. (a) Rainfall, (b) snowfall, (c) actual evaporation, (d) sublimation, (e) blowing snow transport, (f) surface runoff, (g) subsurface flow, (h) groundwater flow, and (i) storage change. Vertical line within the box = median value of the eight-water year data, box = interquartile range (Q1: 25% to Q3: 75%) of the eight-water year data, whiskers = minimum and maximum, circle = mean value of the eight-water year data.



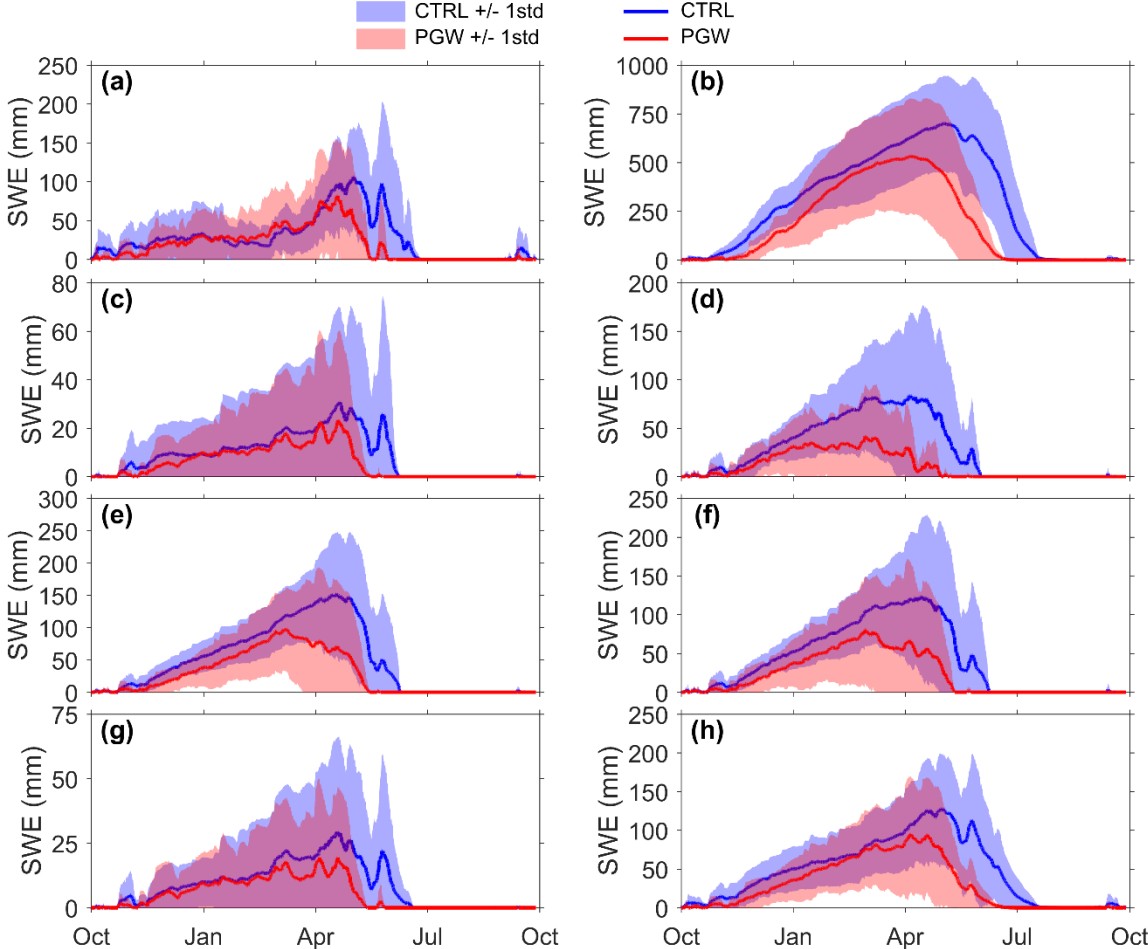

**Figure 7.** Simulated annual mean snow accumulation (SWE) for WRF CTRL and PGW. (a) Alpine, (b) treeline, (c) upper forest, (d) forest clearing blocks, (e) forest circular clearing north-facing, (f) forest circular clearing south-facing, (g) lower forest ecozones, and (h) Marmot Creek basin. Line represents the annual mean and the shadow represents the standard deviation of the eight-water year SWE.





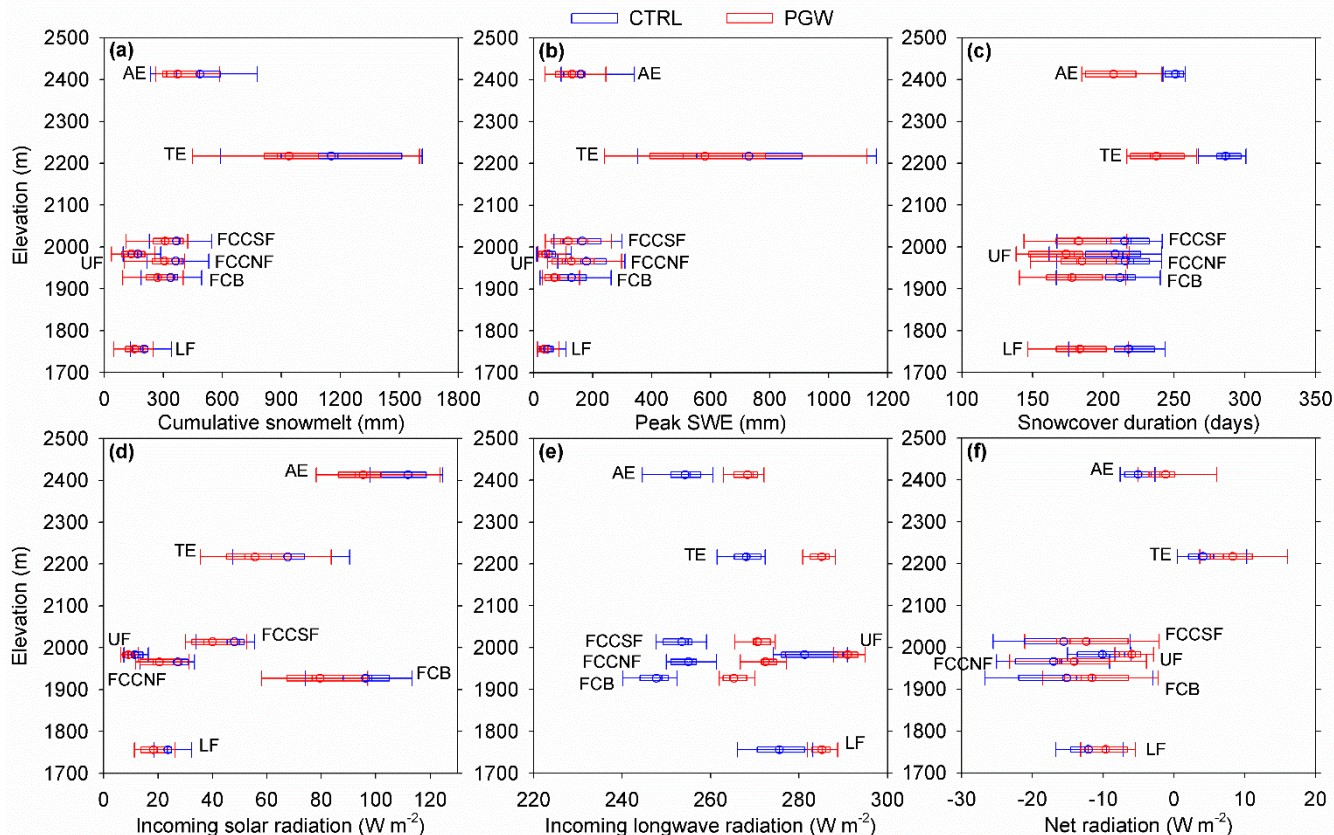

**Figure 8.** Boxplots of the simulated water year (a) cumulative snowmelt, (b) peak snow accumulation (SWE), (c) snowcover duration, (d) incoming solar radiation to snow, (e) incoming longwave radiation to snow, and (f) net radiation to snow for WRF CTRL and PGW for alpine (AE), treeline (TE), upper forest (UF), forest clearing blocks (FCB), forest circular clearing north-facing (FCCNF), forest circular clearing south-facing (FCCSF), lower forest (LF) ecozones. Vertical line within the box = median value of the eight-water year data, box = interquartile range (Q1: 25% to Q3: 75%) of the eight-water year data, whiskers = minimum and maximum, circle = mean value of the eight-water year data.





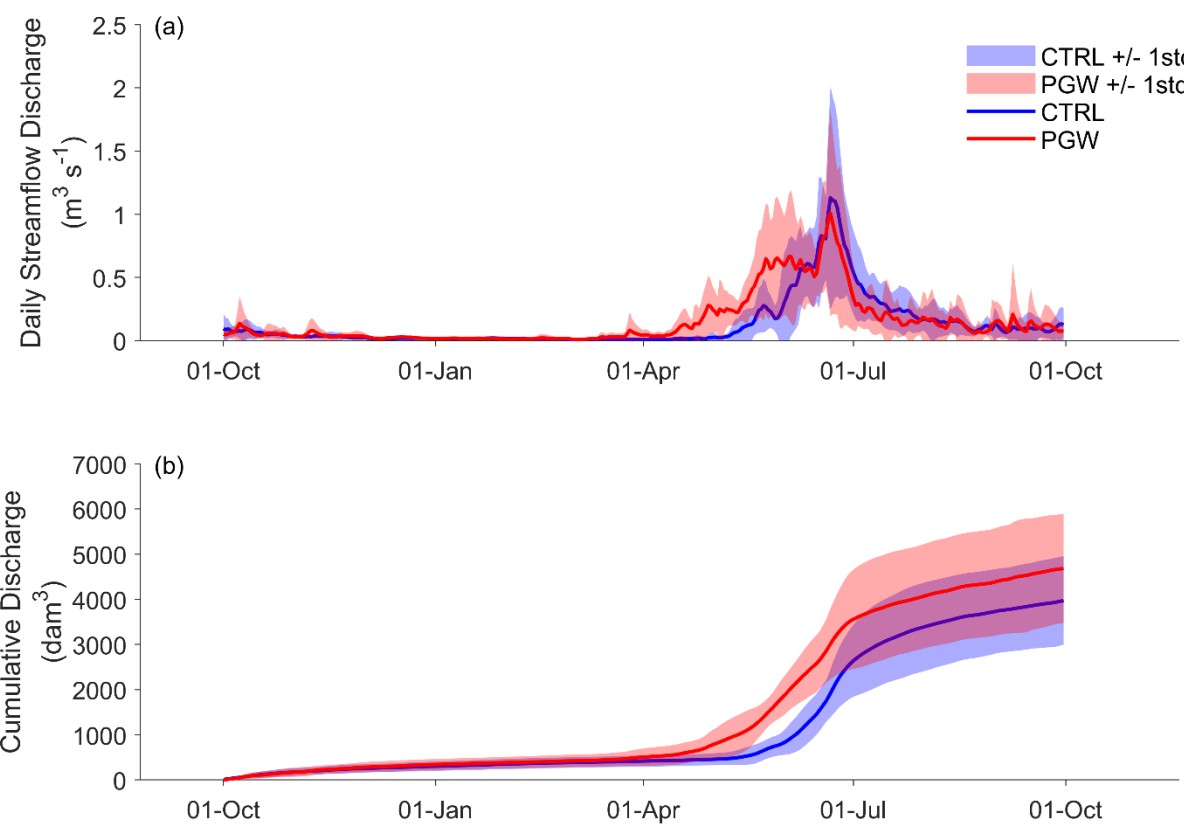

**Figure 9.** Simulated annual mean (a) Marmot Creek daily streamflow discharge and (b) cumulative discharge for WRF CTRL and PGW. Line represents the annual mean and the shadow represents the standard deviation of the eight-water year streamflow discharge.





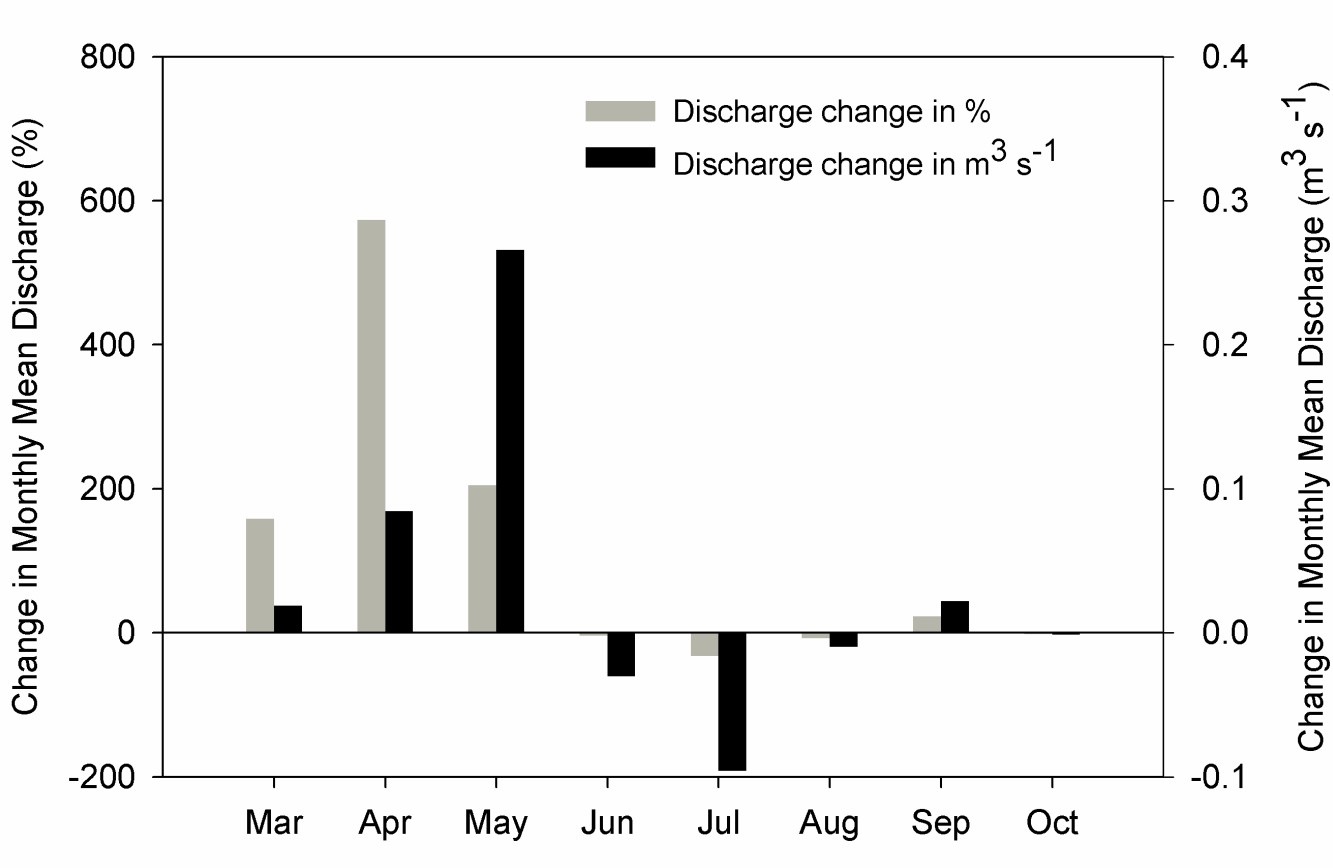

**Figure 10.** Change in the simulated mean Marmot Creek monthly streamflow discharge during March to October for eight-water year between WRF CTRL and PGW.



**Figure 11.** Simulated annual mean daily runoff for WRF CTRL and PGW. (a) Alpine, (b) treeline, (c) upper forest, (d) forest clearing blocks, (e) forest circular clearing north-facing, (f) forest circular clearing south-facing, and (g) lower forest ecozones. Line represents the annual mean and the shadow represents the standard deviation of the eight-water year runoff.





**Figure 12.** Simulated annual mean cumulative runoff for WRF CTRL and PGW. (a) Alpine, (b) treeline, (c) upper forest, (d) forest clearing blocks, (e) forest circular clearing north-facing, (f) forest circular clearing south-facing, and (g) lower forest ecozones. Line represents the annual mean and the shadow represents the standard deviation of the eight-water year runoff.







**Figure 13.** Scatter plots of the mean rainfall ratio and runoff efficiency for WRF CTRL and PGW for alpine (AE), treeline (TE), upper forest (UF), forest clearing blocks (FCB), forest circular clearing north-facing (FCCNF), forest circular clearing south-facing (FCCSF), lower forest (LF) ecozones and Marmot Creek basin.



**Table 1.** Area and mean elevation, aspect, and slope for ecozones at the Marmot Creek Research Basin. Note that the aspect is in degree clockwise from North.

| Ecozone | Area (km$^2$) | Elevation (m a.s.l.) | Aspect (°) | Slope (°) |
|---|---|---|---|---|
| Alpine | 3.23 | 2413 | 110 | 30 |
| Treeline | 0.93 | 2217 | 91 | 22 |
| Upper Forest | 2.75 | 1983 | 108 | 20 |
| Forest Clearing Blocks | 0.40 | 1927 | 140 | 11 |
| Forest Circular Clearing North-facing | 0.26 | 1966 | 34 | 17 |
| Forest Circular Clearing South-facing | 0.24 | 2014 | 113 | 21 |
| Lower Forest | 1.42 | 1756 | 113 | 14 |





**Table 2.** Comparison of observations and WRF outputs for MCRB stations, with mean absolute difference (MAD) and root mean square difference (RMSD). Values are for bias corrected WRF outputs; values inside parentheses are for WRF outputs without bias correction.

| | MAD | | | | | RMSD | | | | |
| --- | --- | --- | --- | --- | --- | --- | --- | --- | --- | --- |
| | Centennial Ridge | Fisera Ridge | Vista View | Upper Clearing | Upper Forest | Centennial Ridge | Fisera Ridge | Vista View | Upper Clearing | Upper Forest |
| Air temperature (°C) | 0.00 | 0.00 | 0.00 | 0.00 | 0.00 | 3.46 | 3.29 | 2.90 | 3.19 | 3.17 |
| | (2.35) | (1.24) | (0.83) | (0.59) | (0.025) | (4.67) | (4.01) | (3.28) | (3.37) | (3.38) |
| Relative humidity (%) | 0.00 | 0.00 | 0.00 | 0.00 | 0.00 | 17.54 | 18.37 | 18.38 | 18.72 | 17.75 |
| | (20.79) | (26.74) | (26.21) | (21.82) | (19.57) | (33.14) | (37.17) | (36.29) | (33.94) | (32.81) |
| Wind speed (m s$^{-1}$) | 0.00 | 0.00 | 0.00 | 0.00 | 0.00 | 4.88 | 2.73 | 0.86 | 0.66 | 0.18 |
| | (1.97) | (1.30) | (2.62) | (3.20) | (3.71) | (4.61) | (2.95) | (3.41) | (3.93) | (4.38) |
| Incoming solar radiation (W m$^{-2}$) | 0.00 | 0.00 | | 0.00 | | 108.16 | 121.22 | | 125.08 | |
| | (28.14) | (17.18) | | (28.87) | | (123.33) | (123.16) | | (132.34) | |
| Precipitation (mm) | | 0.00 | | 0.00 | | | 0.60 | | 0.43 | |
| | | (0.053) | | (0.006) | | | (0.56) | | (0.45) | |





**Table 3.** Evaluation of simulated snow accumulation using the root mean square difference (RMSD, mm SWE), normalised RMSD (NRMSD) and model bias (MB) at the Upper Forest/Clearing and Fisera Ridge sites, Marmot Creek Research Basin.

| | Upper Forest/Clearing | | | Fisera Ridge | | |
|---|---|---|---|---|---|---|
| | Spruce Forest | Forest Clearing | Ridge Top | Upper South-facing Slope | Lower South-facing Slope | Larch Forest |
| RMSD | | | | | | |
| 2007/2008 | 41.0 | 76.6 | 60.3 | 170.1 | 358.8 | 386.2 |
| 2008/2009 | 32.2 | 40.4 | 37.4 | 75.5 | 171.0 | 154.3 |
| 2009/2010 | 22.4 | 65.8 | 110.2 | 33.1 | 129.1 | 96.7 |
| 2010/2011 | 78.3 | 33.3 | 148.7 | 81.2 | 193.3 | 332.3 |
| 2011/2012 | 52.3 | 78.4 | 124.7 | 152.9 | 297.5 | 237.9 |
| 2012/2013 | 40.0 | 58.6 | 107.8 | 141.2 | 142.0 | 103.9 |
| All seasons | 46.5 | 66.1 | 106.5 | 126.9 | 244.8 | 260.1 |
| NRMSD | | | | | | |
| 2007/2008 | 1.05 | 0.77 | 0.66 | 0.50 | 0.64 | 0.62 |
| 2008/2009 | 0.80 | 0.50 | 0.29 | 0.28 | 0.43 | 0.38 |
| 2009/2010 | 0.82 | 0.79 | 1.10 | 0.17 | 0.39 | 0.27 |
| 2010/2011 | 0.85 | 0.22 | 0.89 | 0.28 | 0.53 | 0.83 |
| 2011/2012 | 0.78 | 0.54 | 0.49 | 0.39 | 0.49 | 0.36 |
| 2012/2013 | 0.58 | 0.39 | 0.61 | 0.33 | 0.30 | 0.21 |
| All seasons | 0.84 | 0.55 | 0.68 | 0.39 | 0.52 | 0.51 |
| MB | | | | | | |
| 2007/2008 | -0.89 | -0.71 | 0.36 | -0.48 | -0.62 | -0.59 |
| 2008/2009 | -0.78 | -0.26 | 0.16 | -0.22 | -0.37 | -0.28 |
| 2009/2010 | -0.65 | -0.67 | 0.98 | -0.02 | -0.36 | -0.26 |
| 2010/2011 | 0.42 | 0.06 | 0.55 | 0.13 | 0.43 | 0.68 |
| 2011/2012 | -0.65 | -0.50 | -0.46 | -0.38 | -0.47 | -0.35 |
| 2012/2013 | 0.25 | -0.32 | 0.40 | -0.32 | -0.28 | -0.16 |
| All seasons | -0.31 | -0.43 | 0.17 | -0.27 | -0.33 | -0.22 |




**Table 4.** Evaluation of simulated daily mean streamflow discharge for Marmot Creek using Nash-Sutcliffe efficiency (NSE), root mean square difference (RMSD, $m^3 \, s^{-1}$), normalised RMSD (NRMSD) and model bias (MB).

|             | NSE   | RMSD  | NRMSD | MB     |
|-------------|-------|-------|-------|--------|
| 2006        | 0.44  | 0.146 | 0.76  | 0.01   |
| 2007        | 0.64  | 0.175 | 0.58  | -0.36  |
| 2008        | 0.31  | 0.183 | 0.68  | -0.01  |
| 2009        | -0.33 | 0.180 | 0.91  | 0.000  |
| 2010        | 0.32  | 0.156 | 0.76  | 0.44   |
| 2011        | 0.20  | 0.252 | 0.89  | 0.15   |
| 2012        | 0.72  | 0.173 | 0.55  | -0.27  |
| 2013        | 0.30  | 0.364 | 0.96  | 0.18   |
| All seasons | 0.40  | 0.212 | 0.79  | 0.001  |





**Table 5.** Simulated basin-scale mean annual air temperature (°C), relative humidity (%), wind speed (m s⁻¹), mean annual water balance fluxes (mm), mean annual radiation fluxes to snowcover (W m⁻²), mean seasonal cumulative snowmelt (mm), peak snow accumulation (mm), snowcover duration (days) and melt rate (mm day⁻¹) for WRF CTRL and PGW.

| | CTRL | PGW | Change: PGW - CTRL |
|---|---|---|---|
| Air Temperature | 0.2 | 4.9 | 4.7 |
| Relative Humidity | 69.2 | 67.4 | -1.8 |
| Wind Speed | 2.76 | 2.80 | 0.04 |
| Rainfall | 493 | 761 | 268 |
| Snowfall | 464 | 352 | -112 |
| Total Precipitation | 957 | 1113 | 156 |
| Actual Evaporation | 394 | 518 | 124 |
| Sublimation | 158 | 118 | -40 |
| Blowing Snow Transport | -12 | -10 | 2 |
| Surface Runoff | 106 | 193 | 87 |
| Subsurface Flow | 317 | 305 | -12 |
| Groundwater Flow | 26 | 26 | 0 |
| Total Subsurface Storage | 416 | 404 | -12 |
| Incoming Solar Radiation | 69 | 58 | -11 |
| Incoming Longwave Radiation | 263 | 277 | 14 |
| Net Radiation | -7 | -3 | 4 |
| Cumulative Snowmelt Volume | 401 | 317 | -84 |
| Peak Snow Accumulation | 160 | 119 | -40 |
| Snowcover Duration | 287 | 238 | -49 |
| Melt Rate | 1.3 | 1.1 | -0.2 |