# Peer review of "Figure S1. Quantile-quantile plots of observations and WRF CTRL outputs for Centennial Ridge station in MCRB: (a) WRF CTRL and observed air temperature, (b) corrected WRF CTRL and observed air temperature, (c) WRF CTRL and observed relative humidity (d) corrected WRF CTRL and observed relative h"

_Hydrology and Earth System Sciences, 2019_

## Referee Comment (RC1) · Anonymous Referee #1 · 29 Jan 2020

**Diagnosis of future changes in hydrology for a Canadian Rocky Mountain headwater basin**

by Xing Fang and John W. Pomeroy

submitted to HESS

**Referee comments:**

The manuscript presents a climate change study for a small Canadian Rocky Mountain headwater basin. As process understanding has to be developed at the local scale, such regional/local studies are of a high value. The study is presented in a concise way and contributes well to current discussions. Small changes could improve the readability of the paper (see detailed comments).

However, I would strongly recommend to add a discussion on the uncertainty of the hydrological modelling results. The results chapter is full of numbers, partly with a high number of positions after the decimal point suggesting a high accuracy. Depending on the model design and the catchment characteristics, some results of hydrological modelling are more reliable than others. If, e.g., a model does not represent hydrophobic effects and if they play an important role in a catchment, then the model may simulate overall runoff with a satisfying efficiency but the calculated portion of Hortonian overland flow calculated by the model will be less reliable in this case. Thus, for readers not familiar with CRHM, a discussion on the strengths and especially weaknesses of the model concepts and the resulting reliability of the model results would be very valuable.

The comments in detail:

- The paper contains numerous abbreviations (WRF, MCRB, CRHM, CTRL, PGW, QDM, WY,…). A list of abbreviations would improve the readability.

- In the entire paper: please do not use "alpine" and "treeline" as single words, but always in combination with ecozone: "alpine ecozone", "treeline ecozone". This would improve the grammatical correctness of the sentences and the readability.

- I am no native speaker, but to my feeling sometimes articles are missing, e.g.
  Page 2, line 18: of the world,
  Page 2, line 23: two of the most
  Page 3, line 29: in the eastern slopes
  Page 5, line 5/6: the complex mountain terrain
  Page 9, line 23: had a very comparable … value (or had … values)
  Page 11, line 28: the entire basin
  Page 13, line 16: by a combination
  Page 14, line 15: a large elevational gradient
  Page 14, line 18: these changes were result of the interaction

- Page 3, line 19 and page 4, line 20: A model does not permit convective precipitation processes (they are permitted by the atmospheric conditions), but it permits the

representation/simulation/consideration of convective precipitation processes. Please adapt the formulation.

- Page 3, line 20/21: I would delete "to combine …. from CRHM to" as this is also said by "using a dynamically… model" (line 23-24).

- Page 5, line 29: I would include a citation (e.g. Pomeroy et al., 2007) at the first appearance of the model.

- Page 6, line 1: Please explain dynamic networks of HRUs (in the models I have worked with, HRUs are defined for a catchment and remain the same during the whole simulation).

- Page 6, line 12: please give some short information on the June 2013 flood.

- Page 6, line 26 to 28: Most of the content of this sentence is repeated in the next sentence, please streamline this text.

- Page 6/7, line 31/1: "do not appear to be linear distribution" does not sound to be formulated correctly (linearly distributed).

- Page 7: line 6/7: I would suggest to shorten the sentence, for example in this way: "… for the uncorrected WRF outputs, with two exceptions: Values of RMSD…"

- Page 8, line 22: unit is missing: 112 mm

- Page 8, line 26/27: "Sublimation is the total of blowing snow, surface snowpack and forest canopy interception sublimation." This sentence is either grammatically circular (sublimation is sublimation) or – if the last word does not belong to "blowing snow" - physically incorrect as the blowing of snow means a reduction of snow at the windward site, but by snow transport, not by sublimation in its physical sense.

- Page 9/10, line 31-32/1 and page 14, line 21: Regarding the uncertainty associated with hydrological modelling in general and climate projections, I would recommend to give only one position after the decimal point. Doing so means that you partly loose the differences between the CTRL and PGW values – but that means that they seem to be smaller than the uncertainty.

- Page 10, line 23-25: Please explain why the centre of flow volume shifts to an earlier period in PGW, but the peak basin discharge remains unchanged.

- Page 10, line 32: Please consistently use two positions after the decimal point for the discharge values.

- Page 11, line 26 and page 11, line 28: "close" instead of "closed"

- Page 12, line 3: "September" instead of "Septmeber"

- Page 12, line 28: sublimation losses from blowing snow → see comment above

- Figure 2c: Can you please explain the relative humidity values up to 300%?

- Figure 2e/g: The dotted line for "best linear fit" is misunderstanding. It is just the best linear fit for the lower values. For the whole data set, a best linear fit would look different. I would delete this line.

- Figure 4: I would recommend to show the simulation line in light blue instead of dark blue to get a stronger contrast to the black observation line.

- Figure 11 and 12: The differences between the ecozones would appear clearer if you would use a uniform scaling of the y-axis.

---

## Referee Comment (RC2) · Anonymous Referee #2 · 2 Mar 2020

- 6: skip comma
- 9: „Rocky Mountain**s**"
- 16: „during **a** current period (…) and **a** future period (…)"
- 17: what does „PGW" stand for (comes on page 3)?
- 22: „as well as a shorter snow season": repetition; „in **the** alpine"

- 3: „Empirical snow modelling methods … have great difficulty": reformulate, because it is not the method having a difficulty but we ourselves in interpreting its results with respect to a certain research question

- 32: correct „gird" to „grid"

- 9: „**the** systematic bias"
- 19: „to force **the** hydrological simulations"

- 15: „for **the** current period"
- 16: „for **the** future period"
- 17 and 18: „in **the** CTRL period"
- 20: „calculated by **the** equations"

- 1: better „linearly distributed" than „linear distribution"
- 16/17: „the simulation had large differences with the observations": find better formulation
- 29: „suggesting that **the** model had some"

- 22: „112 **mm**"
- 30: „**the** entire basin"

- in general: always add „**ecozone**" to its name (everywhere)

- 10: „for **the** basin"
- 27: explain „**dam3**"
- 32: delete „Whilst", or connect sentence with previous one („… in May, whilst monthly …")

Page 11:
- 1: „and had **a** very similar low value"
- 7: „onset of **the** spring freshet"
- 9: delete „While", or connect sentence with previous one („… (Fig. 11a), while the largest …")
- 11: „ranging **from** 0.3 mm"
- 13: „from **the** forest clearing"

- 19: „and **the** entire basin"

- 11: „these … simulations **require**" (not „requires", it is plural)
- 9-13: evtl. add Warscher et al. 2019, same (and very recent) discussion
- 18: „in **the** Rocky Mountain**s** region" … and probably better skip „in this region"
- 19: „**One** study suggests …"
- 20: „while other**s** reported"
- 23: „gradient**s**"
- 27: „in **the** PGW period"
- 30: you may even add „for other sites with different climates"
- 31: „of **the** snowcover"

- 5: „in **the** PGW period"
- 12: „for **the** upper forest ecozone"
- 31/32: evtl. add Strasser et al. 2019

- 2: „for **a** more comprehensive"
- 15: „large elevational gradient**s**" or „**a** large elevational gradient"

Fig 1
- „Hydrometeorological Stations" should be „Hydrometeorological Station" (Singular)
- „WRF Grid" should be „WRF Grid centroid", and the latter should be explained in the caption. The „and" before it should be removed

Fig. 2
- caption: „… that **the** best linear fit is **a** straight line …"

Fig. 3
- caption: Comparison without **-s** (better singular). Add what belongs to where in the caption.

Fig. 5
- caption: „Note that **the** total water year precipitation is presented **here,** and **the** average water year value is presented for **the** other variables."

Fig. 6
- caption: „… (FCCSF) **and** lower forest …"

Fig. 7
- caption: better „mean annual"; „**The** line …"

Fig. 9
- caption: „**The** line …"

Fig. 10
- caption: „for **the** eight- water year **period** between …"

Fig. 11
- caption: „**The** line …"

Fig. 12
- caption: „**The** line …"

Fig. 13
- caption: „**Mean rainfall ration and runoff** …   ecozones and **the entire** Marmot Creek basin"

Table 4
- caption: better „mean daily"

Warscher, M., Wagner, S., Marke, T., Laux, P., Strasser, U. and Kunstmann, H. (2019): A 5 km Resolution Regional Climate Simulation for Central Europe: Performance in High Mountain Areas and Seasonal, Regional and Elevation-dependent Variations, Atmosphere, 10, 682, https://doi.org/10.3390/atmos10110682.

Strasser, U., Förster, K., Formayer, H., Marke, T., Meißl, G., Nadeem, I., Stotten, R. and Schermer, M. (2019): Storylines of combined future landuse and climate scenarios and their hydrological impacts in an alpine catchment (Brixental/Austria), Sci. Tot. Env., https://doi.org/10.1016/j.scitotenv.2018.12.077.

---

## Author Response (AR1)

**Response to referees' comments on "Diagnosis of future changes in hydrology for a Canadian Rocky Mountain headwater basin" by Xing Fang and John W. Pomeroy**

**1. Point to point response to review:**

5 **Response to referee #1's comments:**

General comments

The manuscript presents a climate change study for a small Canadian Rocky Mountain headwater basin. As process understanding has to be developed at the local scale, such regional/local studies are of a high
10 value. The study is presented in a concise way and contributes well to current discussions. Small changes could improve the readability of the paper (see detailed comments).

However, I would strongly recommend to add a discussion on the uncertainty of the hydrological modelling results. The results chapter is full of numbers, partly with a high number of positions after the decimal point suggesting a high accuracy. Depending on the model design and the catchment
15 characteristics, some results of hydrological modelling are more reliable than others. If, e.g., a model does not represent hydrophobic effects and if they play an important role in a catchment, then the model may simulate overall runoff with a satisfying efficiency but the calculated portion of Hortonian overland flow calculated by the model will be less reliable in this case. Thus, for readers not familiar with CRHM, a discussion on the strengths and especially weaknesses of the model concepts and the resulting reliability
20 of the model results would be very valuable.

Response to general comments: Thanks to referee #1 for general comments about the manuscript. We added a discussion on the strengths and weakness of model in Discussion section. We also revised the manuscript to improve its structure, readability and flow.

25 The comments in detail:

Detailed comment 1: The paper contains numerous abbreviations (WRF, MCRB, CRHM, CTRL, PGW, QDM, WY,…). A list of abbreviations would improve the readability.

Response 1: Yes, we added an appendix to include a list of abbreviations to improve the readability.

Detailed comment 2: In the entire paper: please do not use "alpine" and "treeline" as single words, but
30 always in combination with ecozone: "alpine ecozone", "treeline ecozone". This would improve the grammatical correctness of the sentences and the readability.

Response 2: Yes, we added ecozone after "alpine" and "treeline" in the revised manuscript.

Detailed comment 3: I am no native speaker, but to my feeling sometimes articles are missing, e.g.

Page 2, line 18: of the world,

Page 2, line 23: two of the most

Page 3, line 29: in the eastern slopes

Page 5, line 5/6: the complex mountain terrain

Page 9, line 23: had a very comparable … value (or had … values)

Page 11, line 28: the entire basin

Page 13, line 16: by a combination

Page 14, line 15: a large elevational gradient

Page 14, line 18: these changes were result of the interaction

Response 3: Yes, we added these missing articles in the revised manuscript.

Detailed comment 4: Page 3, line 19 and page 4, line 20: A model does not permit convective precipitation processes (they are permitted by the atmospheric conditions), but it permits the representation/simulation/consideration of convective precipitation processes. Please adapt the formulation.

Response 4: Yes, we adapted the formulation suggested by the referee.

Detailed comment 5: Page 3, line 20/21: I would delete "to combine …. from CRHM to" as this is also said by "using a dynamically… model" (line 23-24).

Response 5: Yes, we removed the redundant words and rewrote the objectives as: "The objectives of this paper are to use CRHM driven by WRF to: (1) evaluate the ability to simulate snowpack and streamflow regimes in a Canadian Rockies headwater basin without calibration; (2) diagnose the detailed changes in hydrology due to impending climate change for this headwater basin.  By relying on physically based, uncalibrated simulations and dynamical downscaling, it is hoped that the approach introduces a highly robust method for evaluating the impacts of climate change on mountain hydrology."

Detailed comment 6: Page 5, line 29: I would include a citation (e.g. Pomeroy et al., 2007) at the first appearance of the model.

Response 6: Yes, we added Pomeroy et al., 2007 at the first appearance of the model.

Detailed comment 7: Page 6, line 1: Please explain dynamic networks of HRUs (in the models I have worked with, HRUs are defined for a catchment and remain the same during the whole simulation).

Response 7: Yes, HRUs in this model also remain the same throughout the whole simulation, so we deleted "dynamic networks of" to be clearer and reduce confusion.

Detailed comment 8: Page 6, line 12: please give some short information on the June 2013 flood.

Response 8: Yes, we added some brief information for the June 2013 flood: "…the updated model was evaluated in the June 2013 flood when approximately 250 mm precipitation fell at MCRB during 17-24 June 2013 (Fang and Pomeroy, 2016; Pomeroy et al., 2016).".

Detailed comment 9: Page 6, line 26 to 28: Most of the content of this sentence is repeated in the next sentence, please streamline this text.

Response 9: Yes, we combined two sentences to improve the readability and rewrote it as: "Near-surface hourly air temperature, relative humidity, wind speed, precipitation, and shortwave irradiance from observations, uncorrected WRF CTRL outputs, and bias corrected WRF CTRL outputs were compared for the Centennial Ridge, Fisera Ridge, Vista View, Upper Clearing and Upper Forest stations in MCRB, and the comparisons are shown in the quantile-quantile (Q-Q) plots (Fig. 3).".

Detailed comment 10: Page 6/7, line 31/1: "do not appear to be linear distribution" does not sound to be formulated correctly (linearly distributed).

Response 10: Yes, we changed it to "…do not appear to be linearly distributed.".

Detailed comment 11: Page 7: line 6/7: I would suggest to shorten the sentence, for example in this way: "… for the uncorrected WRF outputs, with two exceptions: Values of RMSD…"

Response 11: Yes, we shortened the sentence as suggested by the referee.

Detailed comment 12: Page 8, line 22: unit is missing: 112 mm

Response 12: Yes, we added the missing unit.

Detailed comment 13: Page 8, line 26/27: "Sublimation is the total of blowing snow, surface snowpack and forest canopy interception sublimation." This sentence is either grammatically circular (sublimation is sublimation) or – if the last word does not belong to "blowing snow" - physically incorrect as the blowing of snow means a reduction of snow at the windward site, but by snow transport, not by sublimation in its physical sense.

Response 13: Yes, we changed the wording to make more grammatically sound: "Sublimation is the total flux of snow sublimated from surface snowpack and during blowing snow and forest canopy interception processes…".

Detailed comment 14: Page 9/10, line 31-32/1 and page 14, line 21: Regarding the uncertainty associated with hydrological modelling in general and climate projections, I would recommend to give

only one position after the decimal point. Doing so means that you partly loose the differences between the CTRL and PGW values – but that means that they seem to be smaller than the uncertainty.

Response 14: Yes, we accepted the suggestion and made changes.

"…melt rate was slightly higher for alpine ecozone (i.e. from 1.9 mm day-1 in CTRL to 2.0 mm day-1 in PGW) and remained unchanged for forests ecozones (i.e. 0.6 mm day-1 at upper forest and 0.5 mm day-1 at lower forest in both CTRL and PGW).".

"…snowmelt rates declined by 1.1 mm day-1 for treeline ecozones and by 0.9 to 1.6 mm day-1 for forest clearings ecozones, but increased by 0.1 mm day-1 for alpine ecozone,…"

Detailed comment 15: Page 10, line 23-25: Please explain why the centre of flow volume shifts to an earlier period in PGW, but the peak basin discharge remains unchanged.

Response 15: The centre of flow volume measures the 50% of water year flow volume; its shifting to an earlier period in PGW is caused by a combination of earlier snowpack deletion and snowmelt runoff occurrence in springtime and higher evapotranspiration and consequently lower flow in summertime in PGW. For the peak basin discharge, its timing remains unchanged in PGW, while it is at lower value in PGW. The peak basin discharge is balanced by runoff in all ecozones and is primarily influenced by alpine and treeline ecozones at MCRB. While there is no change in peak runoff date for alpine, forest circular clearing north-facing ecozones, peak runoff occurs four days earlier in treeline ecozone but delays by one day to 9 days in other ecozones. This is a complex streamflow generation system with interplay of hydrological fluxes and states from many processes, and in this case, the changes in peak runoff date in all ecozones happens to result in no change in peak basin discharge date.

Detailed comment 16: Page 10, line 32: Please consistently use two positions after the decimal point for the discharge values.

Response 16: Yes, for consistency, we used two positions after the decimal point for the discharge values.

Detailed comment 17: Page 11, line 26 and page 11, line 28: "close" instead of "closed"

Response 17: Yes, we changed to "close to".

Detailed comment 18: Page 12, line 3: "September" instead of "Septmeber"

Response 18: Yes, we used the corrected word "September".

Detailed comment 19: Page 12, line 28: sublimation losses from blowing snow –> see comment above

Response 19: Yes, we changed wording to "…even though sublimation losses from blowing snow in the alpine ecozone and intercepted snow in the forested ecozones also decreased." to improve clarity.

Detailed comment 20: Figure 2c: Can you please explain the relative humidity values up to 300%?

Response 20: Yes, the values of RH up to 300% are the converted values based on uncorrected WRF air temperature, specific humidity, and specific pressure outputs, and we showed the values of RH up to 300% to indicate the errors in these uncorrected WRF outputs. We added some clarification for the values of RH up to 300% in the Section 3.1 of revised manuscript.

Detailed comment 21: Figure 2e/g: The dotted line for "best linear fit" is misunderstanding. It is just the best linear fit for the lower values. For the whole data set, a best linear fit would look different. I would delete this line.

Response 21: Yes, we deleted the "best linear fit" line in all plots.

Detailed comment 22: Figure 4: I would recommend to show the simulation line in light blue instead of dark blue to get a stronger contrast to the black observation line.

Response 22: Yes, we changed the simulation line to light blue for better contrast.

Detailed comment 23: Figure 11 and 12: The differences between the ecozones would appear clearer if you would use a uniform scaling of the y-axis.

Response 23: We tried to have a uniform scaling of the y-axis for both Figs. 11 and 12, but the figures turned out showing the differences among the ecozones, but they are not great to show the differences between CTRL and PGW for some ecozones, particularly when their values are small, e.g. Fig. 11c, d, g, and Fig. 12c, d, e, f, g.  We included the changed figures with the uniform scaling of the y-axis below, so we think the original Figs 11 and 12 are probably better ones and will keep the original ones.

[Figure]

Figure 11. Simulated annual mean daily runoff for WRF CTRL and PGW. (a) Alpine, (b) treeline, (c) upper forest, (d) forest clearing blocks, (e) forest circular clearing north-facing, (f) forest circular clearing south-facing, and (g) lower forest ecozones. Line represents the annual mean and the shadow represents the standard deviation of the eight-water year runoff.

[Figure]

Figure 12. Simulated annual mean cumulative runoff for WRF CTRL and PGW. (a) Alpine, (b) treeline, (c) upper forest, (d) forest clearing blocks, (e) forest circular clearing north-facing, (f) forest circular clearing south-facing, and (g) lower forest ecozones. Line represents the annual mean and the shadow represents the standard deviation of the eight-water year runoff.

**Response to referee #2's comments:**

General comments

I do mostly agree with referee #1. The paper is a very valuable contribution to the understanding of climate
5 change effects on the hydrology of small high mountain catchments. Its strength are the modelling tools
that were used to provide a physically and process based analysis, and the clear and well-structured text
of the paper. Very well done. Likewise, I am no native speaker, but there is an issue with the use of articles
all through the text. See my supplement. I would also suggest to always add "ecozone" after its name.
Other things that would improve the overall value of the paper:

10 - a brief explanation of how the ecozones were derived

- some more words about the generation of the PGW simulation (particularly extending "The climate
perturbation was derived from 19-model ensemble mean change from the fifth phase of the Coupled
Model Intercomparison Project (CMIP5; Taylor et al., 2012) under a "business as usual" forcing scenario:
representative concentration pathway 8.5 (RCP8.5; van Vuuren et al., 2011).")

15 - a map of the HRUs

Everything else: see supplementary comments.

Congratulations, a very nice paper, very interesting, and fun to read. Good work!

Response to general comments: Thanks to referee #2 for general comments about the manuscript. We
added a brief explanation of the ecozones generation along with a map of the HRUs, because they are
20 connected: ecozone were derived from the same landcover type of the HRUs. We also added a few more
words for PGW simulation. Last, we added some missing articles as suggested in the detailed
supplementary comments, and we also added "ecozone" after its name. We also revised the manuscript
to improve its structure, readability and flow.

Supplementary comments

25 Supplementary comment 1:

- 6: skip comma

- 9: „Rocky Mountains"

- 16: „during a current period (…) and a future period (…)"

30 - 17: what does „PGW" stand for (comes on page 3)?

- 22: „as well as a shorter snow season": repetition; „in the alpine"

Response 1: We keep the comma for readability although the comma can be skipped. The sentence would be quite long when skipping comma.
Yes, we changed to use "…Canadian Rockies".
Yes, we added the missing articles "a".
Yes, we used "…a current period (2005-2013) and a future pseudo global warming period (PGW, 2091-2099)" to replace "…current period (CTRL, 2005-2013) and future period (PGW, 2091-2099). Also, an appendix for abbrevations is added, which is suggested by the referee #1.
We rephrased to "The alpine snow season will be shortened by almost one and half month, but at some lower elevations there will be large decreases in peak snowpack (~45%) in addition to a shorter snow season."; this is to convey the message "besides a shorter snow season, large decreases in peak snowpack occurred at some lower elevations".

Supplementary comment 2:

- 3: „Empirical snow modelling methods … have great difficulty": reformulate, because it is not the method having a difficulty but we ourselves in interpreting its results with respect to a certain research question

Response 2: Yes, we rephrased to "Empirical snowmelt modelling methods that use temperature-index techniques are inappropriate in cold mountain regions…".

Supplementary comment 3:

- 32: correct „gird" to „grid"

Response 3: Yes, we made correction.

Supplementary comment 4:

- 9: „**the** systematic bias"

- 19: „to force **the** hydrological simulations"

Response 4: Yes, we made the changes as suggested.

Supplementary comment 5:

- 15: „for **the** current period"

- 16: „for **the** future period"

- 17 and 18: „in **the** CTRL period"

- 20: „calculated by **the** equations"

5  Response 5: Yes, we made the changes as suggested.

Supplementary comment 6:

- 1: better „linearly distributed" than „linear distribution"

- 16/17: „the simulation had large differences with the observations": find better

10  formulation

- 29: „suggesting that **the** model had some"

Response 6: Yes, we made the changes as suggested.

Supplementary comment 7:

15  - 22: „112 **mm**"

- 30: „**the** entire basin"

- in general: always add „**ecozone**" to its name (everywhere)

Response 7: Yes, we made the changes as suggested.

Supplementary comment 8:

20  Page 10

- 10: „for **the** basin"

- 27: explain „**dam3**"

- 32: delete „Whilst", or connect sentence with previous one („… in May, whilst

monthly …")

25  Response 8: Yes, we added the missing article "the".
dam3 stands for cubic decametre, equal to 1000m3. It is one of the SI units, and according to journal's

house standards, units do not need to be defined in text.

Yes, we changed it to "…in May. In contrast, monthly…".

Supplementary comment 9:

Page 11:

- 1: „and had **a** very similar low value"

- 7: „onset of **the** spring freshet"

- 9: delete „While", or connect sentence with previous one („… (Fig. 11a), while the

largest …")

- 11: „ranging **from** 0.3 mm"

- 13: „from **the** forest clearing"

- 19: „and **the** entire basin"

Response 9: Yes, we made the changes as suggested.

Supplementary comment 10:

- 11: „these … simulations **require**" (not „requires", it is plural)

- 9-13: evtl. add Warscher et al. 2019, same (and very recent) discussion

- 18: „in **the** Rocky Mountain**s** region" … and probably better skip „in this region"

- 19: „**One** study suggests …"

- 20: „while other**s** reported"

- 23: „gradient**s**"

- 27: „in **the** PGW period"

- 30: you may even add „for other sites with different climates"

- 31: „of **the** snowcover"

Response 10: Yes, we made the changes as suggested and added Warscher et al. 2019 to the discussion.

Supplementary comment 11:

- 5: „in **the** PGW period“

- 12: „for **the** upper forest ecozone“

- 31/32: evtl. add Strasser et al. 2019

Response 11: Yes, we made the changes as suggested and added Strasser et al. 2019 to the discussion.

Supplementary comment 12:

- 2: „for **a** more comprehensive“

- 15: „large elevational gradient**s**“ or „**a** large elevational gradient“

Response 12: Yes, we made the changes as suggested.

Supplementary comment 13:

Fig 1

- „Hydrometeorological Stations“ should be „Hydrometeorological Station“ (Singular)

- „WRF Grid“ should be „WRF Grid centroid“, and the latter should be explained in the caption. The „and“ before it should be removed

Response 13: Yes, we made the changes as suggested.

Supplementary comment 14:

Fig. 2

- caption: „… that **the** best linear fit is **a** straight line …“

Response 14: In the new Fig. 3, we removed the best linear fit as suggested by the referee #1.

Supplementary comment 15:

Fig. 3

- caption: Comparison without **-s** (better singular). Add what belongs to where in the caption.

Response 15: Yes, we changed it to "Comparison of the…".

Supplementary comment 16:

Fig. 5

- caption: „Note that **the** total water year precipitation is presented **here,** and **the** average water year value is presented for **the** other variables."

Response 16: We made change: "…Note that the accumulation over the water year is used for precipitation, and the average value over the water year is presented for the other variables.".

5  Supplementary comment 17:

Fig. 6

- caption: „… (FCCSF) **and** lower forest …"

Response 17: Yes, we made the change as suggested.

Supplementary comment 18:

10  Fig. 7

- caption: better „mean annual"; „**The** line …"

Response 18: Yes, we made the change as suggested.

Supplementary comment 19:

Fig. 9

15  - caption: „**The** line …"

Response 19: Yes, we made the change as suggested.

Supplementary comment 20:

Fig. 10

- caption: „for **the** eight- water year **period** between …"

20  Response 20: Yes, we made the change as suggested. Now "Figure 11. Change between WRF CTRL and PGW periods in the simulated mean Marmot Creek monthly streamflow discharges during March to October for the eight-water years."

Supplementary comment 21:

Fig. 11

25  - caption: „The line …"

Response 21: Yes, we made the change as suggested.

Supplementary comment 22:

Fig. 12

- caption: „The line …"

Response 22: Yes, we made the change as suggested.

Supplementary comment 23:

Fig. 13

- caption: „**Mean rainfall ration and runoff** … ecozones and **the entire** Marmot

Creek basin"

Response 23: Yes, we made the change as suggested.

Supplementary comment 24:

Table 4

- caption: better „mean daily"

Response 24: The evaluation is for the time-series of daily mean streamflow value (i.e. daily discharge) from the simulation. We think daily mean is better one to use.

**2. A list of relevant changes made in the manuscript:**

The following lists the relevant changes in addition to changes made based on reviewers' comments and suggestions.

- Slight change in title to "**Diagnosis of future changes in hydrology for a Canadian Rockies headwater basin**"
- Moved brief introduction of CRHM to **1. Introduction** section from **2.3 Hydrological model and simulations** section. This is to introduce CRHM to readers early on, rather than in method section.
- Edited **4. Discussion** section to improve readability, flow and make arguments better.
- Revised figures in **Supplement**: deleted the "best linear fit" line in all quantile-quantile plots.
- Other edits in English throughout the revised manuscript to improve its readability.

The revised manuscript with marked-up changes is shown in the next.

**Diagnosis of future changes in hydrology for a Canadian Rockiesy Mountain headwater basin**

Xing Fang and John W. Pomeroy

[1]Centre for Hydrology, University of Saskatchewan, Saskatoon, S7N 1K2, Canada

5   *Correspondence to*: Xing Fang (xing.fang@usask.ca)

**Abstract.** Climate change is anticipated to have impacts on the hydrologywater resources of the Saskatchewan River, which originatess in the Canadian Rockiesy Mountains mountain range.  To better understand the climate change impacts in the mountain headwaters of this basin, a physically based hydrological model was developed for this basin using the Cold Regions Hydrological Modelling platform (CRHM) for Marmot Creek Research Basin (~9.4 km$^2$), located in the Front Ranges of the

10  Canadian Rockiesy Mountains.  Marmot Creek is composed of ecozones ranging from montane forests to alpine tundra and alpine exposed rock and includes both large and small clearcuts.  The model included blowing and intercepted snow redistribution, sublimation, energy-balance snowmelt, slope and canopy effects on melt, Penman-Monteith evapotranspiration, infiltration to frozen and unfrozen soils, hillslope hydrology, streamflow routing and groundwater components and was parameterised without calibration from streamflow.   Near-surface outputs from the 4-km Weather Research and Forecasting

15  (WRF) model were bias-corrected using the quantile delta mapping method with respect to meteorological data from five stations located from low elevation montane forests to alpine ridgetopsmountaintop and running overduring October 2005-September 2013.  The bias corrected WRF outputs during a currentcurrent period (CTRL, 2005-2013) and a future pseudo global warmingfuture period (PGW, 2091-2099) were used to drive model simulations to assess changes in Marmot Creek's hydrology.  Under a "business as usual" forcing scenario: representative concentration pathway 8.5 (RCP8.5) in PGW, the

20  basin will warms up by 4.7 °C and receives 16% more precipitation, which will leads to a 40 mm decline in seasonal peak snowpack, 84 mm decrease in snowmelt volume, 0.2 mm day$^{-1}$ slower melt rate, and 49 days shorter snowcover duration.  The alpine snow season will be shortened by almost one and half month, but at some lower elevations there will beare large decreases in peak snowpack (~45%) in addition toas well as a shorter snow season.  Declines in the 
[revised manuscript text omitted]

subalpineother ecozones, the seasonal snowpack declined underwent substantially declines and decreased throughout the season within PGW., and t The date of seasonal snowpack depletion advanced from early August to late June with PGW at the treeline ecozone and from mid-June to late May in the other ecozones. For eight WY, tThe mean snowmelt rate was estimated by dividing mean annual annual peak SWE by the number of days from peak SWE to snowpack disappearaenceepletion and

5  with PGW was lower for the treeline and forest clearings ecozones in PGW, decliningwith decreases ranging by from 0.9 mm day$^{-1}$ inat the forest clearing blocks ecozone (i.e. from 1.4 mm day$^{-1}$ in CTRL to 0.5 mm day$^{-1}$ in PGW) to 1.6 mm day$^{-1}$ inat the forest circular clearing north-facing ecozone (i.e. from 2.9 mm day$^{-1}$ in CTRL to 1.3 mm day$^{-1}$ in PGW). Whilst the melt rate was slightly higher for alpine ecozone (i.e. from 1.9 mm day$^{-1}$ in CTRL to 2.0 mm day$^{-1}$ in PGW) and remained unchanged for forests ecozones, (i.e. 0.6 mm day$^{-1}$ at upper forest and 0.5 with increases ranging from 0.01 mm day$^{-1}$ at upper forest (i.e.

10  from 0.63 mm day$^{-1}$ in CTRL to 0.64 mm day$^{-1}$ in PGW) to 0.04 mm day$^{-1}$ at both alpine (i.e. from 1.95 mm day$^{-1}$ in CTRL to 1.99 mm day$^{-1}$ in PGW) and lower forest in both CTRL and PGW) (i.e. from 0.46 mm day$^{-1}$ in CTRL to 0.50 mm day$^{-1}$ in PGW). For the entire basin, there was a very small decline in the melt rate from 1.3 mm day$^{-1}$ in CTRL to 1.1 mm day$^{-1}$ in PGW (Table 5).

     Changes in the seasonal total snowmelt, peak SWE, snowcover duration and radiation fluxes to snowcover from eight

15  WY were also compared between CTRL and PGW. Figure 98a shows that cumulative snowmelt volume decreased within PGW for all ecozones, and for the eight WY mean seasonaltotal snowmelt at the, treeline ecozone declined the mostsuffered highest decrease by (215 mm), with the decreaseslines elsewhere ranging from 32 mm at upper forest to 113 mm at alpine ecozone. The pPeak SWE declinedof seasonal snowpack reduced for all ecozones within PGW, with the largest and decrease in the mean value of eight WY peak SWE was highest in at the treeline ecozone by (149 mm) and the lowest at the upper and

20  lower forests ecozones by (11 mm), as shown in (Fig. 89b). The duration of seasonal snowcoverpack declined became shorter for all ecozones iwithn PGW, with the eight WY mean snowcover duration shortened, ranging from by 31 days at the forest circular clearing north-facing ecozone to 49 days at the treeline ecozone (Fig. 98c). Table 5 shows that the for the basin-wide, eight WY mean snowmelt volume, peak SWE and duration of seasonal snowcoverpack decreased by 84 mm, 40 mm and 49 days, respectively. The seasonalannual net radiation to snowcover increased slightly within PGW for all ecozones, ranging

25  from an additional 2 W m$^{-2}$ at lower forest to 4 W m$^{-2}$ at other ecozones (Fig. 89f). The increases in the net radiation to snowcover wereas due to because of higher annuallongwave irradiance to snowcover in with PGW for all ecozones, ranging from an additional 10 W m$^{-2}$ tohigher at the lower forest ecozone to 17 W m$^{-2}$ higher atto the treeline and forest clearings ecozones (Fig. 89e), whilst the annual seasonal solar irradiance to snowcover declinedreduced fo for all ecozones with PGW, with declines ranging from 2 W m$^{-2}$ less at the upper forest ecozone to 17 W m$^{-2}$ less at the forest clearing blocks ecozone (Fig.

30  98d). BFor the entire basin-wide, in PGW, annualseasonal solar irradiance to snowcover decreased by 11 W m$^{-2}$ and longwave irradiance to snowcover decreased by 11 W m$^{-2}$ and increased by 14 W m$^{-2}$, respectively, resulting in an increase ofwith 4 W m$^{-2}$ increase in annual net radiation to snowcover with PGW (Table 5).

**3.6 Changes in streamflow**

Simulated daily streamflow discharge was compared  between CTRL  and PGW  in Fig 10a, which shows the annual time-series of Marmot Creek  discharge for CTRL and PGW periods. The results suggest and illustrates th the basin discharge was

5 very similar  over winter through mid-March. However, the average onset of spring freshet advanced by 45 days. from 8 May for CTRL to 24 March with PGW, with the centre of flow volume occurring 12 days earlier from 22 June for CTRL to 10 June with PGW. The peak basin discharge was 1.13 $m^3 s^{-1}$ and 1.01 $m^3 s^{-1}$ in CTRL and withCompared to CTRL, the ddeclinedinlimbbetween t~~

10 the date of peak discharge  through to late August declined with PGW compared to the CTRL period,  because of higher evaporation losses with PGW  (Fig. 7c). Despite the lower peak and faster recession, The cumulative annual discharge volume increased with PGW by 18% from 3973 $dam^3$ in CTRL to 4683 $dam^3$ with PGW (Fig. 10b).  The change in  monthly discharge was calculated by subtracting the monthly discharge with PGW

15 from that in  CTRL period. Figure 11 shows noticeable increases in monthly discharge  with PGW for  March to May and September, with PGW ise ranging from 0.02 $m^3 s^{-1}$ in March to 0.27 $m^3 s^{-1}$ in May compared to CTRL monthly discharge of 0.01 $m^3 s^{-1}$ in March and 0.13 $m^3 s^{-1}$ in May. In contrast. Whilst monthly discharge within PGW declined notably in June and July by 0.03 $m^3 s^{-1}$ and 0.09 $m^3 s^{-1}$ from  CTRL values of 0.69 $m^3 s^{-1}$  and 0.29 $m^3 s^{-1}$ , respectively. Monthly discharge in PGW decreased by only 0.01 $m^3 s^{-1}$ in August

20 from the CTRL value of 0.13 $m^3 s^{-1}$  and was  similar low values  (0.06 $m^3 s^{-1}$   in October. Monthly fractional changs in  discharge with PGW ranged from a 573% increase in April to a 33% decrease in July seasonal fractional changes in discharge ranged from a   236% increase in spring  ( March to May), a 12% decline in summer ( June to August) and a 13% increase in early  fall ( September to October).

25 The simulated daily runoff fluxes (surface, sub-surface and groundwater runoff) and annual runoff volumes were  plotted for all ecozones in MCR  to examine  changes between CTRL and PGW. Figures 12-12 consistently show minimal change in winter months and an advance in the onset of  spring freshet for all ecozones with PGW. The annual peak runoff from the alpine ecozone decreased from 25.6 mm day$^{-1}$ in CTRL to 23.2 mm day$^{-1}$ with PGW, both occurring on 20 June (Fig. 12a). Whilst the greatest decline in annual peak runoff occurred from the treeline

30 ecozone, from 27.8 mm day$^{-1}$ in CTRL to 19.5 mm day$^{-1}$ with PGW, on 21 June and 17 June, respectively (Fig. 12b). There were moderate declines in annual peak runoff with PGW from other ecozones, ranging from 0.3 mm day$^{-1}$ less at the lower forest ecozone to 2.0 mm day$^{-1}$ less at the forest circular clearing north-facing ecozone. The change in   dates of annual peak runoff with PGW ranged from no change at the forest clearing north-facing ecozone to  9

days later at the forest clearing south-facing ecozone (Figs. 121c-g).  There were moderate increases in the annual runoff volumes from the forest clearing and lower forest ecozones within PGW, ranging from 12 dam$^3$ increase fromat the lower forest ecozone to a 17 dam$^3$ increase fromat the forest clearing blocks ecozones (Figs. 132d-g).  The annual runoff volume from the upper forest ecozone increased from 258 dam$^3$ in CTRL to 316 dam$^3$ within PGW (Fig. 132c).  The For alpine and treeline ecozones are the, primary sources for Marmot Creek basin discharge; here, the annual runoff volume increased 25%substantially, from the alpine ecozone from 2457 dam$^3$ in CTRL to 3065 dam$^3$ within PGW, from the alpine ecozone, (i.e. about 25% increase) but decreased 2% from the treeline ecozone from 1007 dam$^3$ in CTRL to 986 dam$^3$ within PGW from the treeline ecozone (i.e. about 2% decline) (Figs. 123a-b).

The relationship between rainfall ratio (RR) and runoff efficiency (RE) wasere examined for all ecozones and the Marmot Creek entire basin forin CTRL and PGW periods.  The A RRrainfall ratio is defined asthe cumulative total rainfall divided by cumulative total precipitation overfor a WYwater year, and REa runoff efficiency is the cumulativedefined as total runoff (surface, subsurface and groundwater runoff) divided by cumulative total precipitation for a WYwater year.  A RR > 0.5 indicates a rainfall-dominated precipitation regime, and a RR < 0.5 indicates a snowfall-dominated precipitation regime.  The RE describes how well the basin or ecozone convertsthe fraction of precipitation volumes to that is transformed to runoff volumes by different ecozones in the basin and it normally varies between 1 and 0.  Figure 134 illustrates the changes between CTRL and PGW in mean values of RR and RE values for all ecozones and the whole basin between CTRL and PGW.  The mean RR in CTRL was 0.43 for the alpine ecozone, meaning it is snowfall-dominated, and that for treeline ecozone wais 0.51, closed to an equal snowfall and rainfall precipitation regime.  For other ecozones, the mean RR in CTRL were between 0.6 and 0.62, indicating rainfall dominance in CTRL.  With PGWIn contrast, the mean RR increased in PGW and to ranged from 0.61 at the alpine ecozone to 0.76 at the lower forest ecozone. F, and for the entire basin, the mean RR rose from 0.52 in CTRL (i.e. closed to equal snowfall and rainfall) to 0.68 within PGW (i.e. rainfall-dominated).  The mean RE stayed relatively unchanged for the forest ecozones, ranging from 0.1 for lower forest ecozone to 0.11 for upper forest ecozone in both CTRL and PGW periods.  For the forest clearing ecozones, mean RE values dropped by 0.02.  The mean RE changed substantially had large changes in the alpine and treeline ecozones; it droppinged from 1.04 in CTRL to 0.91 within PGW for the treeline ecozone but increasinged from 0.62 in CTRL to 0.69 in PGW for the alpine ecozone.  The value near 1 in CTRL for treelines ecozones refers to melt of late lying snow patches.  For the entire basin, the RE mean SGE increased by only 0.01 with PGW, from 0.44 in CTRL to 0.45 in PGW despite the basin shifting towards domination by rainfall runoff.

**4 Discussion**

[revised manuscript text omitted]